# Anti-fibrotic activity of a rho-kinase inhibitor restores outflow function and intraocular pressure homeostasis

Guorong Li[1], Chanyoung Lee[2], A Thomas Read[2], Ke Wang[2], Jungmin Ha[3], Megan Kuhn[1], Iris Navarro[1], Jenny Cui[1], Katherine Young[2], Rahul Gorijavolu[1], Todd Sulchek[3], Casey Kopczynski[4], Sina Farsiu[1,5], John Samples[6], Pratap Challa[1], C Ross Ethier[2,3]*, W Daniel Stamer[1,5]*

[1]Department of Ophthalmology, Duke University, Durham, United States; [2]Department of Biomedical Engineering, Georgia Institute of Technology/Emory University, Atlanta, United States; [3]Department of Mechanical Engineering, Georgia Institute of Technology, Atlanta, United States; [4]Aerie Pharmaceuticals, Inc, Durham, United States; [5]Department of Biomedical Engineering, Duke University, Durham, United States; [6]Washington State University Floyd Elson School of Medicine, Spokane, United States

*For correspondence:
ross.ethier@bme.gatech.edu (CRE);
dan.stamer@duke.edu (WDS)

**Abstract** Glucocorticoids are widely used as an ophthalmic medication. A common, sight-threatening adverse event of glucocorticoid usage is ocular hypertension, caused by dysfunction of the conventional outflow pathway. We report that netarsudil, a rho-kinase inhibitor, decreased glucocorticoid-induced ocular hypertension in patients whose intraocular pressures were poorly controlled by standard medications. Mechanistic studies in our established mouse model of glucocorticoid-induced ocular hypertension show that netarsudil both prevented and reduced intraocular pressure elevation. Further, netarsudil attenuated characteristic steroid-induced pathologies as assessed by quantification of outflow function and tissue stiffness, and morphological and immunohistochemical indicators of tissue fibrosis. Thus, rho-kinase inhibitors act directly on conventional outflow cells to prevent or attenuate fibrotic disease processes in glucocorticoid-induced ocular hypertension in an immune-privileged environment. Moreover, these data motivate the need for a randomized prospective clinical study to determine whether netarsudil is indeed superior to first-line anti-glaucoma drugs in lowering steroid-induced ocular hypertension.

## Introduction

Topical glucocorticoids (GCs) are routinely used after ocular surgery and to treat common ocular disorders such as uveitis and macular edema (*Noble and Goa, 1998*; *Haeck et al., 2011*). Unfortunately, ocular hypertension (OHT, i.e. elevated intraocular pressure [IOP]) is a common adverse event of such treatment, which can progress to a form of secondary glaucoma known as steroid-induced glaucoma. Interestingly, 90% of people with primary open-angle glaucoma (POAG), the most common type of glaucoma, develop OHT after GC treatment (*Becker, 1965*). This is more than double the rate of the general population, likely because GCs increase extracellular matrix (ECM) material, cell contractility, and stiffness in an already dysfunctional conventional outflow pathway (*Johnson et al., 1997*; *Zhou et al., 1998*), as is found in POAG (*Rönkkö et al., 2007*; *Tamm and Fuchshofer, 2007*). The conventional outflow pathway drains the majority of the aqueous humor, and its hydrodynamic resistance is the primary determinant and homeostatic regulator of

IOP (*Tamm, 2009*). Thus, pro-fibrotic changes in this outflow pathway, such as occurring with GC treatment, frequently lead to significantly elevated IOPs (*Liu et al., 2018*).

Rho-associated protein kinase (ROCK) is a major cytoskeletal regulator in health and disease. Indeed, ROCK mediates pro-fibrotic processes in many tissues and pathological conditions, including fibrotic kidney disease (*Musso et al., 2017*), idiopathic pulmonary fibrosis (*Marinković et al., 2013*; *Knipe et al., 2015*), cardiac fibrosis (*Olson, 2008*; *Haudek et al., 2009*), liver fibrosis, intestinal fibrotic strictures associated with Crohn's disease (*Crespi et al., 2020*), vitreoretinal disease (*Yamaguchi et al., 2017*), and lens capsule opacity (*Korol et al., 2016*). Hence, rho-kinase inhibitors (ROCKi) have been evaluated as anti-fibrotic therapeutics in multiple contexts. However, the mechanisms underlying their anti-fibrotic activity are complex and multifactorial due to the central involvement of ROCK in many cellular processes. Thus, tissue/pathology-specific studies are essential to evaluate the efficacy of ROCKis as anti-fibrotic agents.

Mechanistic studies evaluating anti-fibrotic activity of ROCKi in the eye are particularly interesting in view of the eye's immune-privileged status (*Streilein, 2003*), which minimizes the role of macrophages and monocytes as compared to other organ systems. In fact, OHT in glaucoma patients is the only approved indication for ROCKi in humans thus far (*Roskoski, 2020*). Specifically, two ROCKi, ripasudil and netarsudil (NT), have been approved for clinical use to treat OHT in glaucoma patients because of their safety profile and IOP-lowering ability and ultimately their unique mechanism of action (*Garnock-Jones, 2014*; *Isobe et al., 2014*; *Tanihara et al., 2015*; *Kopczynski and Heah, 2018*; *Serle et al., 2018*). Therefore, ROCKi are the only available glaucoma drug class that directly targets and improves conventional outflow function (*Schehlein and Robin, 2019*). However, their use is currently limited to patients whose IOPs are inadequately controlled by medications that target other ocular sites. We here show that local delivery of the ROCKi, NT, significantly lowers IOP in two cohorts of steroid-induced glaucoma patients refractory to conventional medications. Moreover, we help define NT's mechanism of restorative, anti-fibrotic action in treating steroid-induced OHT using our established mouse model (*Li et al., 2019*).

## Results

### NT lowered IOP in steroid-induced glaucoma patients whose OHT was poorly controlled by standard glaucoma medications

Based on changes to the trabecular meshwork (TM) previously observed in studies of steroid glaucoma (*Johnson et al., 1997*; *Overby et al., 2014*; *Li et al., 2019*), we tested NT's efficacy at lowering IOP in steroid-induced ocular hypertensive patients who did not respond well to standard first-line treatments. We retrospectively reviewed patient records, forming two cohorts of subjects from three treatment locations. The first cohort was created by an unbiased retrospective search of the Duke Eye Center's electronic medical records using the key words, 'steroid responder/glaucoma' and 'netarsudil'. Our search identified 21 eyes of 19 patients (mean age 66.8 years), treated with GCs for a variety of ocular conditions and who demonstrated OHT secondary to GC treatment (*Table 1*, cohort 1). In accordance with current standard of treatment, these steroid-responsive patients were initially treated with aqueous humor suppressants (e.g., carbonic anhydrase inhibitors and/or adrenergics, *Table 1*). We note that prostaglandin analogues were typically used as second-line therapies (or not at all) in this patient cohort, due to concerns about the possible pro-inflammatory effects of these agents (*Table 1*). Unfortunately, these first- and second-line medications did not adequately control IOP in these patients, who presented with an average IOP of $24.3 \pm 6.6$ mmHg before NT treatment (mean $\pm$ SD). Thus, NT treatment was initiated, resulting in a clinically significant lowering of IOP in all patients within 1 month, presenting an average IOP decrease of 7.9 mmHg (p=$1.2\times10^{-7}$, *Figure 1A*). In this cohort, IOP was reduced to an average of $16.4 \pm 4.9$ mmHg, which is within the normal range. IOPs over a 3-month course of treatment for each patient are shown in *Figure 1—figure supplement 1*. It is important to note that none of the patients were 'tapered' from their steroid during the first month of treatment, and thus the observed IOP lowering cannot be due to a removal of steroid.

To evaluate NT efficacy in a second cohort of patients, a retrospective chart review was conducted on all patients seen by an experienced glaucoma specialist (JRS) over a 1-month period at two geographic locations. This cohort included patients with a single diagnosis of steroid-induced

**Table 1.** Netarsudil (NT) effectively lowered steroid-induced ocular hypertension in patients whose IOP was not well controlled with standard glaucoma medications.

| Patient/ eye | Age (years) | Gender | Race | Ocular condition | Steroid type | Glaucoma medications | IOP before NT (mmHg) | IOP < 1 month after NT (mmHg) |
|---|---|---|---|---|---|---|---|---|
| Cohort 1 | | | | | | | | |
| 1/OD | 50 | M | AA | POAG | Durezol | Brim, Cos | 23 | 16 |
| 2/OD | 74 | F | AA | CACG | PF | Apra, Meth, Lat | 33 | 22 |
| 3/OD | 27 | M | HIS | TG | PF | Bet, Brim, Dor; Lum | 36 | 28 |
| 4/OD | 66 | F | CAU | Uveitis | Retisert | Cos, Brim | 17 | 9 |
| 5/OS | 76 | F | CAU | POAG | PF | Lum, Cos | 17 | 11 |
| 6/OS | 53 | M | CAU | SR | Ozurdex | Com, Dor | 31 | 12 |
| 7/OS | 74 | F | CAU | SR | PF | Cos, Brim | 25 | 16 |
| 8/OS | 61 | F | AA | Uveitis | Durezol | Cos, Brim, Lat | 34 | 26 |
| 9/OD | 81 | M | UNK | POAG | PF | Cos, Brim, Lat | 25 | 19 |
| 9/OS | | | | | | | 20 | 16 |
| 10/OS | 51 | F | CAU | Uveitis; SR | PF; Fluti | Cos, Brim | 26 | 15 |
| 11/OS | 78 | F | CAU | POAG; SR | Oral Pred | Lat, Tim, Dor, Dmx | 22 | 12 |
| 12/OD | 76 | F | CAU | SIG | Lotemax | Brim, Cos, Vyzulta | 28 | 20 |
| 12/OS | | | | | | | 27 | 19 |
| 13/OD | 62 | F | CAU | SIG | PF | Tim, Dor, Lat | 21 | 15 |
| 14/OD | 78 | M | AA | POAG; SR | PF | Com, Dor, Lat | 30 | 13 |
| 15/OS | 72 | F | CAU | LTG; SIG | PF | Brim, Lat | 15 | 13 |
| 16/OS | 83 | F | AA | Glaucoma | PF; Retisert | Com, Lat | 21 | 18 |
| 17/OD | 76 | M | CAU | POAG; SIG | Durezol | Cos, Brim | 12 | 10 |
| 18/OS | 83 | F | CAU | POAG; SIG | PF | Cos, Brim | 29 | 18 |
| 19/OD | 49 | F | CAU | POAG; SIG | Lotemax | Cos, Brim | 18 | 17 |
| | 66.8 | 6M/13F | | | | 2.7 ± 0.7 | 24.3 ± 6.6 | 16.4 ± 4.9 |
| Cohort 2 | | | | | | | | |
| 20/OS | 60 | M | CAU | Uveitis | Ivt. Dex | Com, Lat | 60 | 44 |
| 21/OS | 92 | F | CAU | SR | Ozurdex | Lat, Alph, Dor, Tim | 17 | 12 |
| 22/OD | 77 | F | CAU | SR | Oral Pred | Lat | 24 | 19 |
| 22/OS | | | | | | | 24 | 18 |
| 23/OD | 65 | F | CAU | SR | Ozurdex | Lum | 30 | 26 |
| 23/OS | | | | | | | 33 | 22 |
| 24/OD | 16 | F | CAU | PCG | Topical Pred | Tim, Azo, Lat | 21 | 16 |
| 25/OD | 67 | M | CAU | POAG | Topical Pred | Lat, Azo, Bet | 28 | 18 |
| 26/OS | 3d | M | CAU | POAG | Oral Pred | Tim, Lat, Alph | 44 | 32 |
| 27/OD | 90 | M | CAU | POAG | Topical Pred | Azo, Lat | 22 | 16 |
| 27/OS | | | | | | | 22 | 16 |
| | 58.4 | 4M/4F | | | | 2.4 ± 1.1 | 26.5 ± 7.7 | 19.5 ± 5.8 |

IOP, intraocular pressure; OD, right eye; OS, left eye; M, male; F, female; AA, African American; HIS, Hispanic; CAU, Caucasian; UNK, unknown; POAG, primary open-angle glaucoma; CACG, chronic angle-closure glaucoma; TG, traumatic glaucoma; OHT, ocular hypertension; PCG primary congenital glaucoma; PF=Pred Forte; Dex=Dexamethasone; Pred=Prednisone; Ivt=intravitreal; Brim=Brimonidine; Com=Combigan; Cos=Cosopt; Apra=Apraclonidine; Meth=Methazolamide; Lat=Latanoprost, Bet=Betaxolol; Dor=Dorzolamide; Lum=Lumigan; Com=Combigan; Alph=Alphagan; Tim=Timolol; Azo=Azopt. IOPs of patient 10/OS were statistically determined to be outliers and the numbers are removed from the analysis.

glaucoma that was uncontrolled on standard glaucoma medications, and who subsequently received NT. We identified 11 eyes from eight patients, with an average age of 58.4 years (*Table 1*, Cohort 2). The IOP in these eyes prior to NT treatment was similar to the first cohort at 26.5 ± 7.7 mmHg (mean ± SD). As well, NT significantly lowered IOP in this second cohort by an average of 7.0 mmHg (p=0.0003, *Figure 1B*), leading to an IOP of 19.5 ± 5.8 mmHg (mean ± SD). We conclude that NT significantly lowers IOP in steroid glaucoma patients who were refractory to conventional anti-ocular hypertensive medications.

## Increase in outflow facility by NT effectively prevents and reverses steroid-induced OHT in a mouse model of human disease

To better understand the mechanism of NT's IOP-lowering effect in patients, we carried out studies utilizing our established steroid-induced OHT mouse model (*Li et al., 2019*). It is known that daily treatment with NT significantly decreases IOP in naïve mouse eyes by improving conventional outflow function (*Li et al., 2016*). Thus, we studied the efficacy of NT treatment in mice, focusing specifically on NT's effect on conventional aqueous outflow dynamics and TM structure and function. Our experimental studies were designed to address two clinically important questions in this mouse model: (1) Can NT prevent steroid-induced OHT? and (2) can NT reverse steroid-induced OHT?

To test for prevention, mice were treated unilaterally with either NT or placebo (PL) starting 1 day prior to bilateral delivery of dexamethasone (DEX)-loaded nanoparticles (NPs), with NT or PL treatment continuing for the 4-week duration of DEX-NP exposure. Baseline IOPs in both NT and PL groups were similar (19.4 ± 0.4 and 19.6 ± 1.1 mmHg, respectively, p=0.60). One day after NT or PL treatment, but before DEX treatment, IOP was 20.1 ± 0.9 mmHg in PL-treated eyes and significantly lower (16.1 ± 1.4 mmHg) in NT-treated eyes (p=0.0009, *Figure 2A*). IOPs were followed for 4 weeks, and average IOP elevation in PL-treated eyes was 5.94 ± 0.57 mmHg, while IOP in NT-treated eyes was significantly lower than IOP in PL eyes (p<0.0001, *Figure 2B*), returning close to baseline levels (average IOP elevation compared to baseline of 0.23 ± 0.45 mmHg, p=0.71). Outflow facility, the primary determinant of IOP, was 84% greater in NT eyes compared to PL (4.87 ± 1.09 vs. 2.64 ± 0.44 nl/min/mmHg, p=0.08, *Figure 2C*). Long-term treatment with NT appeared to impact outflow facility in both treated and contralateral eyes (4.87 vs. 4.39 nl/min/mmHg, respectively), likely due to contralateral effects as observed previously with DEX (*Li et al., 2019*).

Motivated by these findings, we next tested NT's ability to reverse steroid-induced OHT. We delivered DEX-NPs bilaterally for 3 weeks to mice *before* initiating unilateral NT or PL therapy (once per day for 4 consecutive days). The baseline IOPs before DEX treatment in both NT and PL groups were similar (18.5 ± 1.76 vs. 19.2 ± 1.28 mmHg, respectively, p=0.21). After 3 days of DEX-NP administration, IOP was significantly elevated in both groups (*Figure 2D*). After 3 weeks of DEX-NP treatment, average IOP elevation (days 3–21) compared to baseline in PL and NT cohorts was similar (6.77 ± 0.67 vs. 7.74 ± 0.384 mmHg, respectively, p=0.45, *Figure 2E*). Commencement of NT treatment resulted in rapid IOP lowering (within 1 day) followed by a continued decrease. After 4 days of treatment, the change in IOP from baseline in PL-treated eyes was 8.19 ± 0.46 vs. 2.69 ± 0.47 mmHg for NT-treated eyes (p<$10^{-4}$). In fact, NT almost completely reversed steroid-induced OHT, returning IOP to near baseline levels (18.5 ± 1.76 vs. 21.4 ± 1.30 mmHg). NT increased outflow facility by 33% compared to PL (5.18 ± 0.57 vs. 3.48 ± 0.51 nl/min/mmHg, p=0.038, *Figure 2F*) and increased outflow by 37% compared to contralateral eyes (3.27 ± 0.49 nl/min/ mmHg, p=0.025). Assuming no effect of NT on aqueous inflow rate, these measured facility differences can mathematically account for 93–102% of the observed IOP differences between NT- and PL-treated eyes, both here and in the prevention study. If a 10–15% reduction of aqueous inflow is accounted for, the measured facility differences account for 85–95% of the observed IOP changes. We conclude that NT's IOP-lowering effect in this steroid glaucoma model can be largely explained by increased outflow facility.

## Reduction of steroid-induced TM stiffening by NT

Mechanical stiffness of the TM, a key tissue in the conventional outflow pathway, was shown to be increased by GC treatment, negatively correlating with outflow facility (*Wang et al., 2018*). We also previously showed that GC treatment decreased the tendency of the Schlemm's canal (SC) lumen to collapse at elevated IOPs (*Li et al., 2019*), an effect that appeared to be mediated by changes in

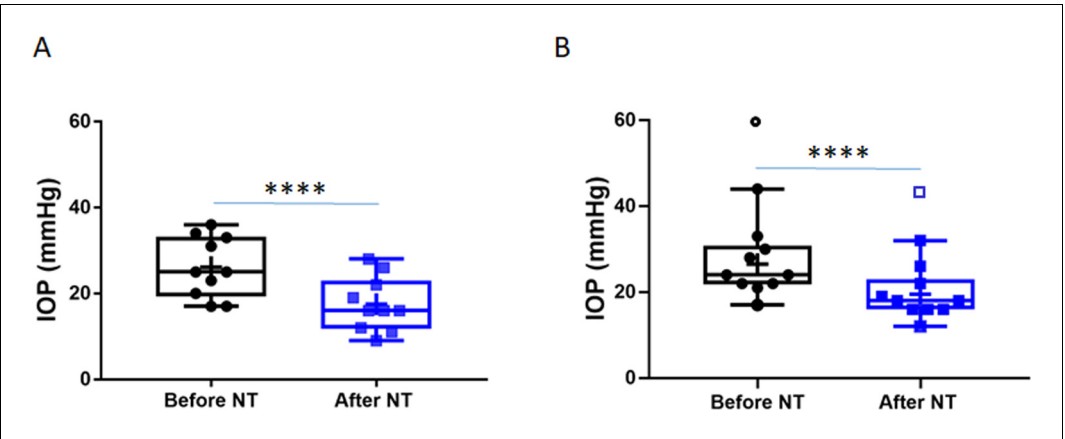

**Figure 1.** Netarsudil (NT) efficaciously lowered intraocular pressure (IOP) in patients with steroid-induced elevated IOP poorly controlled with standard glaucoma medications. IOPs were measured by Goldmann applanation tonometry in patients who demonstrated ocular hypertension (OHT) after steroid treatment for a variety of ocular conditions (*Table 1*). These individuals were initially treated with aqueous humor suppressants and/or prostaglandin analogues but showed persistent OHT, and were thus treated with NT. (A) Shows IOPs of patient cohort 1 (n = 21 eyes of 19 patients) and (B) Shows IOPs of patient cohort 2 (n = 11 eyes of 8 patients). 'Before NT' indicates IOPs measured before initiating NT in these patients, and 'after NT' shows IOP after daily treatment with NT for 1 month or less. The central line in box and whisker represents the median, the top and bottom edges are 25th and 75th percentiles, respectively, the whiskers extend to the most extreme data points, and '+' indicates the mean. Empty symbols were statistically determined to be outliers of data sets. ****p<0.0001.

The online version of this article includes the following figure supplement(s) for figure 1:

**Figure supplement 1.** Time course of netarsudil (NT) effects on intraocular pressure (IOP) in patients with steroid-induced elevated IOP poorly controlled with standard glaucoma medications.

TM stiffness. Since ROCKi such as NT decrease cellular contractility and act as anti-fibrotic agents (*Lin et al., 2018*), we hypothesized that NT would reverse steroid-induced conventional outflow tissue stiffening and lead to more SC collapse at elevated IOPs. To test this hypothesis, we used our reversal protocol as above. On day 5 after the last NT/PL treatment, mice were anesthetized and secured on a custom imaging platform (*Li et al., 2014a*; *Li et al., 2014b*; *Boussommier-Calleja et al., 2015*; *Li et al., 2016*; *Li et al., 2019*). The anterior chamber of the NT- or PL-treated eye was cannulated with a single needle connected to a fluid reservoir, allowing IOP to be controlled. The conventional outflow tissues were imaged using optical coherence tomography (OCT) as IOP was clamped at five different levels. With increasing IOP, the SC lumen became smaller in both NT- and PL-treated eyes, but to different extents. In PL-treated eyes, the SC lumen was still patent at an IOP of 20 mmHg, while the SC lumen in NT-treated eyes was almost completely collapsed. In fact, the SC luminal cross-sectional areas differed significantly between NT- and PL-treated groups at each pressure level (p=0.0024, *Figure 3* and *Figure 3—figure supplement 1*).

We next quantified NT effects on TM stiffness using inverse finite element modeling (iFEM) (*Li et al., 2019*) based on the OCT images acquired in vivo. This iFEM procedure, allowing us to deduce TM stiffness based on structural analysis of SC collapse and associated TM deformation, was performed on OCT images from both NT- and PL-treated groups. We deduced TM tissue stiffness values of 22 kPa in the NT-treated group vs. 61 kPa in the PL-treated group (*Figure 3G*). The latter value is consistent with the TM stiffness we previously measured in DEX-NP treated eyes of 69 kPa (*Li et al., 2019*). Notably, the TM stiffness we estimated in NT-treated eyes was close to the value of 29 kPa that we previously determined in naïve eyes (*Li et al., 2019*; *Figure 3—figure supplement 2*). Thus, it appears that NT returns TM stiffness to near control levels after only 4 days of administration in eyes made hypertensive by long-term GC administration.

It was desirable to obtain an independent and direct measurement of TM stiffness in a cohort of NT-treated mouse eyes. For this purpose, we used atomic force microscopy (AFM) (*Wang et al., 2017b*; *Wang et al., 2018*; *Li et al., 2019*). Using the reversal treatment protocol, we observed that NT significantly reduced TM tissue stiffness by 23% compared to contralateral eyes (1.54 ± 0.19 vs.

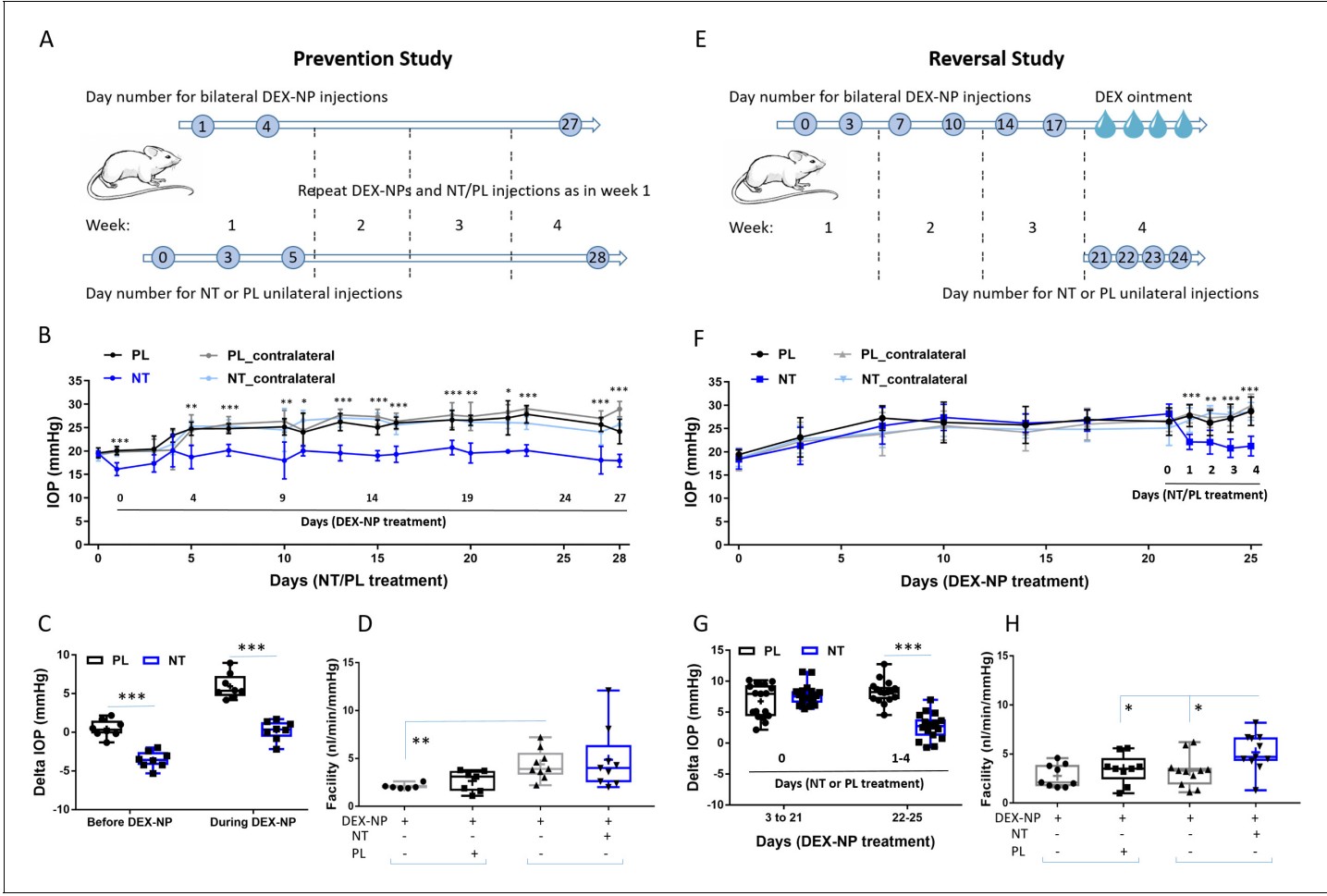

**Figure 2.** Netarsudil (NT) prevented and reversed steroid-induced ocular hypertension by improving outflow function. (**A**) In the prevention study, intraocular pressure (IOP) was measured in two groups of age- and gender-matched wild-type C57BL/6 mice receiving NT or placebo (PL) unilaterally by subconjunctival injections. The day after NT/PL treatment was started, dexamethasone-loaded nanoparticles (DEX-NPs) were delivered bilaterally into the periocular space to release DEX and create steroid ocular hypertension. We continued to apply NT or PL 2–3 times per week to the same eye and to deliver DEX-NPs 1–2 times per week bilaterally for 4 weeks. (**B**) The IOP data over time are displayed as mean ± SD (n = 8 for each group). *p<0.05, **p<0.01, ***p<0.001 comparing NT vs. PL groups. (**C**) We summarize the data from Panel B by showing the average IOP elevation over baseline ('delta IOP') following NT or PL treatment in the presence of continuous DEX-NP delivery. 'Before DEX-NP' refers to IOP elevations at 1 day post-NT or -PL treatment but before delivery of DEX-NPs, and 'during DEX-NP' refers to IOP elevations averaged over 4–28 days of NT or PL treatment, i.e., over 3–27 days of DEX-NP delivery. ***p<0.001 comparing NT vs. PL groups. (**D**) After 4 weeks of PL or NT treatment, outflow facility was measured in freshly enucleated eyes (p=0.08 for comparison of DEX-NP+PL vs. DEX-NP+NT, n = 7–9). Brackets indicate paired eyes. (**E**) In the second (reversal) study, IOP was measured in two groups of age- and gender-matched mice receiving DEX-NPs bilaterally 1–2 times per week for 3 weeks. During the last 5 days, NT or PL was administered unilaterally and DEX ointment bilaterally, once per day for four consecutive days. (**F**) We found that NT rapidly reversed three weeks of DEX-NP-induced ocular hypertension. The data show mean ± SD (n = 17 for PL group and n = 19 for NT group). **p<0.01, ***p<0.001 comparing NT vs. PL groups. (**G**) We summarize the data from Panel F by showing average IOP elevations above baseline ('delta IOP') for DEX-NP-treated eyes in both groups prior to NT or PL treatment (left side) or averaged over 1–4 days of NT or PL treatment (right side). ***p<0.001 comparing NT vs. PL groups. (**H**) On day 5 of NT/PL treatment, both eyes were enucleated and outflow facility was measured (*p=0.038, n = 9 for DEX-DP+PL and n = 11 for DEX-NP+NT group). Brackets indicate paired eyes. See **Figure 1** for interpretation of box and whisker plots. Source data for IOPs and outflow facilities in **Figure 2—source data 1**.

The online version of this article includes the following source data and figure supplement(s) for figure 2:

**Source data 1.** Source data for IOPs and outflow facilities in **Figure 2**.

**Figure supplement 1.** Netarsudil (NT) reversed dexamethasone-loaded nanoparticle (DEX-NP)-induced intraocular pressure (IOP) elevation even after 3 months.

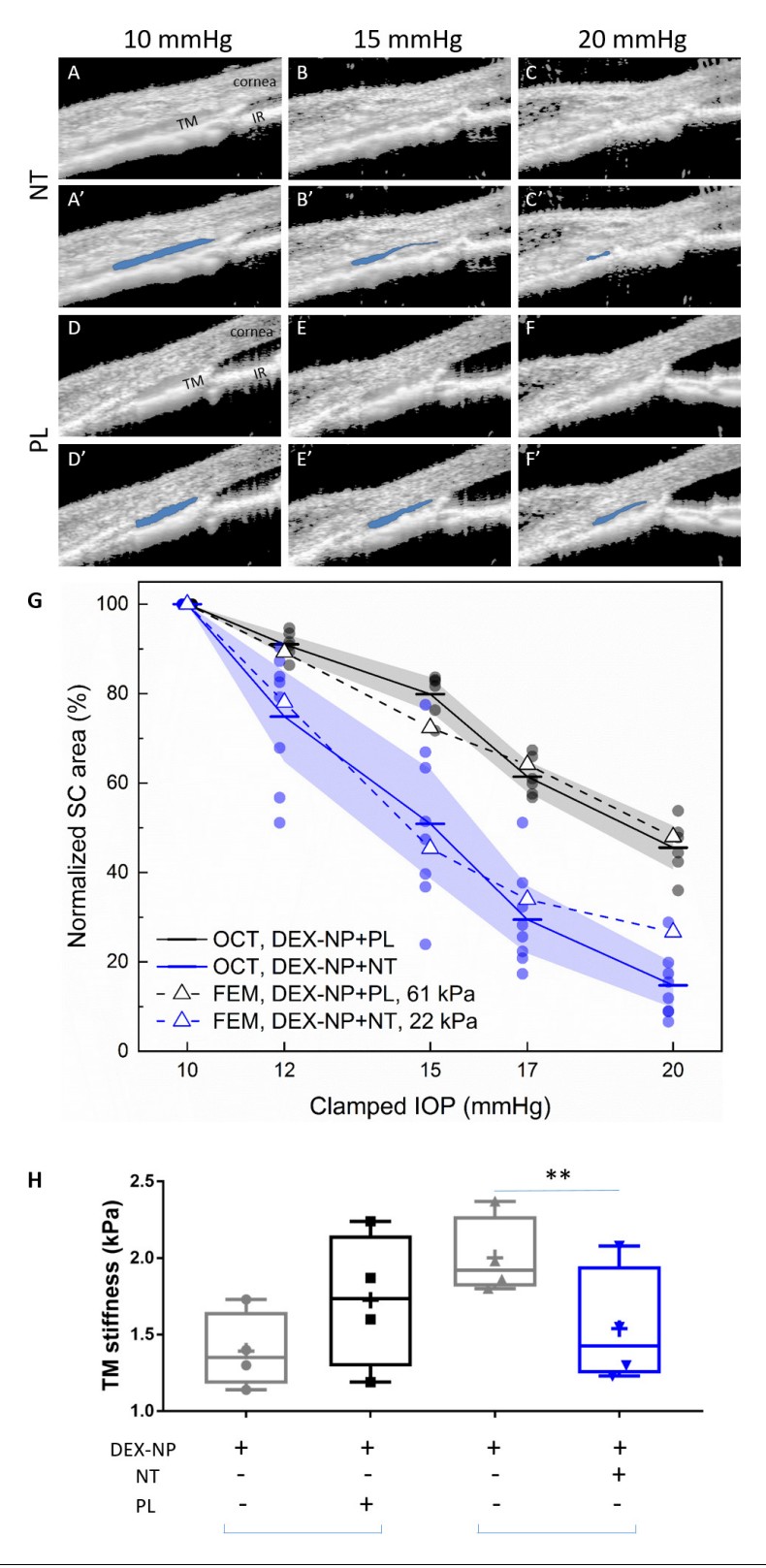

**Figure 3.** Netarsudil (NT) reduced steroid-induced trabecular meshwork (TM) stiffening visualized in living mice by spectral domain-optical coherence tomography (SD-OCT) and estimated by inverse finite element modeling (iFEM) and measured directly by atomic force microscopy (AFM). Wild-type C57BL/6 mice received dexamethasone-loaded nanoparticles (DEX-NPs) bilaterally for 4 weeks. During the last week, mice received either NT or placebo (PL) unilaterally for 4 consecutive days. (A–F) On day 5, living mouse eyes were cannulated to control intraocular pressure (IOP) and were

*Figure 3 continued on next page*

*Figure 3 continued*

subjected to sequentially increasing pressure steps (10–20 mmHg) while imaging conventional outflow tissues using OCT. Images were analyzed and Schlemm's canal (SC) lumen semi-automatically segmented (in blue) using custom SchlemmSeg software (A'–F'). The increased tendency toward SC collapse in NT- vs. PL-treated eyes is evident. (G) Quantitative comparison of SC lumen areas (blue regions in Panels A'-F') in both NT and PL treatment group at five clamped IOPs (10, 12, 15, 17, and 20 mmHg). The plotted quantity is relative SC area (normalized to value at 10 mmHg), and the data show an increased tendency toward SC collapse in NT-treated eyes compared to PL-treated eyes. The dots indicate individual eyes, and bars represent mean values for each IOP. Shaded regions indicate 95% confidence intervals. N = 6 for PL and n = 8 for NT treatment groups. We conducted iFEM to structurally analyze the response of anterior segment tissues to varying IOP levels, mimicking the experimental measurements. Dashed lines show results of the iFEM analysis for least squares best fit TM stiffness values, yielding estimated stiffnesses of 61 kPa for PL-treated eyes and 22 kPa for NT-treated eyes. Abbreviations: IR = iris. (H) At day 5, eyes were collected and processed for atomic force measurements of TM stiffness (Young's modulus). We observed that NT softened the TM vs. control eyes. Brackets indicate paired eyes. We point out that the magnitudes of the AFM and OCT/iFEM measurements of TM stiffness differ due to the well-known dependence of soft tissue stiffness on loading mode (compression by AFM vs. tension by SC lumen pressurization) (*Ethier and Simmons, 2007*), as discussed in more detail in *Wang et al., 2017a*. An average of 135 measurements were made from three quadrants on four biological samples for both cohorts, p=0.007. Blue brackets indicate paired eyes. See *Figure 1* for interpretation of box and whisker plots. Source data for area of SC lumen in *Figure 3—source data 1*. Source data for FEM in *Figure 3—source data 2*. Source data for AFM measurements in *Figure 3—source data 3*.

The online version of this article includes the following source data and figure supplement(s) for figure 3:

**Source data 1.** Source data for segmentation of area of SC lumen in *Figure 3*.
**Source data 2.** Source data for FEM in *Figure 3*.
**Source data 3.** Source data for AFM measurements in *Figure 3*.
**Figure supplement 1.** Netarsudil (NT) enhanced intraocular pressure (IOP)-induced collapse of Schlemm's canal (SC) lumen in ocular hypertensive eyes as visualized by spectral domain-optical coherence tomography (SD-OCT).
**Figure supplement 2.** Quantitative comparison of normalized Schlemm's canal (SC) lumen areas in netarsudil (NT) and placebo (PL) treatment groups at five clamped intraocular pressures (IOPs) (10, 12, 15, 17, and 20 mmHg).
**Figure supplement 3.** A representative cryosection showing the limbal region during AFM measurements of TM stiffness, acquired on a tissue sample immersed in PBS.

---

$2.00 \pm 0.13$ kPa, p=0.007, *Figure 3H*). In contrast, PL did not affect TM tissue stiffness compared to contralateral eyes ($1.39 \pm 0.12$ vs. $1.73 \pm 0.20$ kPa, p=0.39). When stiffness measurements of PL and NT groups were normalized to untreated contralateral eyes, a trend toward NT-induced softening was observed but was not significant perhaps due to small sample size (p=0.15).

## Anti-fibrotic activity of NT in GC-treated conventional outflow tissues

We next investigated NT effects on fibrotic and morphological changes in TM tissues. At the light microscopic level, we observed no gross morphological changes in conventional outflow tissues in NT- vs. PL-treated steroid-induced OHT eyes (*Figure 4A–C*). In contrast, we observed that NT treatment reduced the expression of alpha smooth muscle actin (αSMA; *Figure 4H* vs. 4I) and fibronectin (FN; *Figure 4E* vs. 4F), two fibrotic indicators known to be elevated in conventional outflow tissues after GC treatment. Mean fluorescence intensities from the TM region of sections in PL vs. NT treatment groups were quantified in a masked fashion (see *Figure 4—figure supplement 1* for an example). Results indicate that NT treatment significantly reduced both αSMA (4K, p=0.037) and FN protein expression (4J, p=0.007). In fact, NT-mediated down-regulation led to αSMA and FN levels similar to those observed previously in eyes treated with phosphate buffered saline (PBS)-loaded (sham) NPs (*Li et al., 2019*) and naïve eyes (4D and 4G).

When examined at the electron microscopic level, we observed two major effects of NT treatment on steroid-induced OHT eyes (*Figure 5*). The first was a significant reduction in the amount and density of basement membrane materials (BMM) underlying the inner wall of SC. The second was an apparent increase in the number of 'open spaces' in the TM of NT-treated eyes, particularly in the juxtacanalicular (JCT) region. These two changes were scored on a semi-quantitative scale, confirming observational impressions of NT treatment (*Figure 5F*, p=0.02). In addition, we quantified the amount of BMM below the inner wall of SC in each treatment group following the Lütjen-Drecoll approach (*Figure 5—figure supplement 1*; *Overby et al., 2014*). Similar to results reported in Overby et al., we found that length of BMM underlying SC was significantly increased in mice after DEX treatment (*Figure 5G*, p<0.001). Indeed, our BMM ratio measurements in control and DEX-treated mice were quantitatively similar to those reported by Overby and colleagues. Most

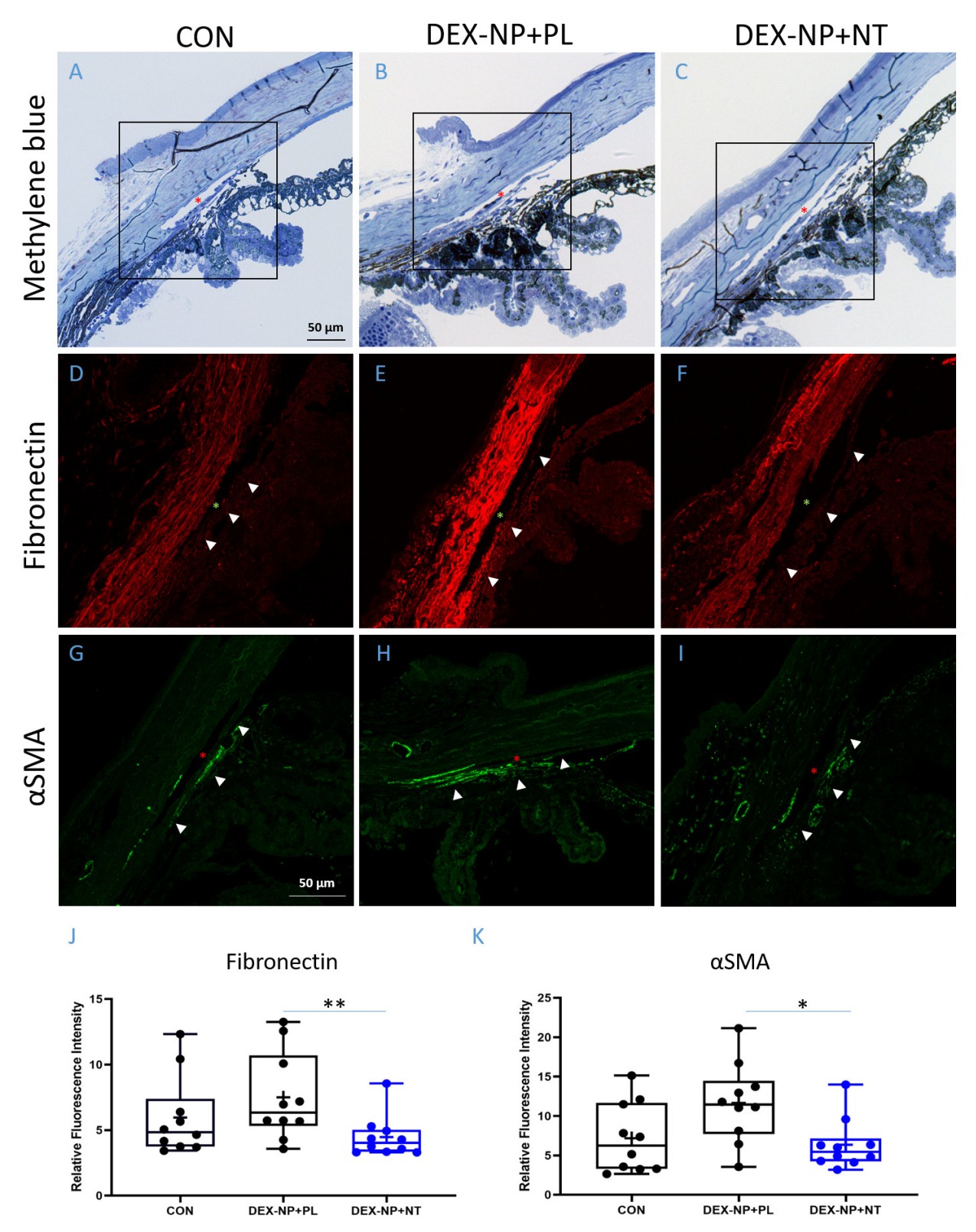

**Figure 4.** Netarsudil (NT) decreased fibrotic markers in ocular hypertensive mouse eyes. Two groups of age- and gender-matched wild-type C57BL/6 mice had dexamethasone-loaded nanoparticles (DEX-NPs) delivered bilaterally 1–2 times per week for 4 weeks to create ocular hypertension. During the last week of DEX-NP exposure, mice were split into two subgroups and treated unilaterally with either NT or placebo (PL) once per day for 4 consecutive days. At day 5, eyes were collected and processed for histological analysis and compared to untreated control eyes. (A–C) Representative

*Figure 4 continued on next page*

*Figure 4 continued*

sagittal sections of iridocorneal tissues visualized by light microscopy after methylene blue staining, showing normal gross morphology in NT-treated eyes. Boxes in A–C indicate areas of interest displayed in Panels D–I. Control, PL- or NT-treated ocular hypertensive eyes were sectioned and iridocorneal tissues were probed with antibodies recognizing fibronectin (FN) (D–F) or alpha-smooth muscle actin (αSMA; G–I). Identical confocal settings were used for all samples, which were imaged on the same day. Data shown are representative of samples from 10 images from seven control mice, and 10 images from five mice treated with NT or PL. Nuclei were counterstained with DAPI in (C–F). Asterisks indicate Schlemm's canal (SC) lumen, arrowheads show trabecular meshwork. (J, K) Fluorescence intensity from the trabecular meshwork region of each image was quantified in a masked fashion (*Figure 4—figure supplement 1*) as summarized in box and whisker plots. Data show average fluorescence intensity measurements of trabecular meshwork (TM) from different sections of eyes under different treatment conditions. See *Figure 1* for interpretation of box and whisker plots. n = 5 eyes for PL- and NT-treated eyes, n = 7 for untreated eyes (CON). *p<0.05 and **p<0.01. Source data for quantification of FN and αSMA fluorescence in TM in *Figure 4—source data 1* and *Figure 4—source data 2*, respectively.

The online version of this article includes the following source data and figure supplement(s) for figure 4:

**Source data 1.** Source data for quantification of FN in *Figure 4*.
**Source data 2.** Source dat for quantification of SMA in *Figure 4*.
**Figure supplement 1.** Quantitative analysis of fibronectin (FN) and alpha-smooth muscle actin (αSMA) expression in trabecular meshwork (TM).

importantly, these data confirm significant differences between PL- and NT-treated eyes (*Figure 5G*, p=0.001), whereby NT appears to partially restore the ultrastructure of the subendothelial region.

## Discussion

The major finding of the current study was that NT effectively decreased steroid-induced OHT in two different cohorts of patients who were refractory to standard glaucoma medications. This clinical observation was mechanistically studied in a reliable, established mouse model of steroid-induced OHT (*Li et al., 2019*). As in patients, NT decreased long-term steroid-induced OHT in mice and also prevented the onset of steroid-induced OHT. NT-mediated rescue of OHT in mice was accompanied by restoration of normal outflow facility and conventional outflow tissue stiffness, as well as significant morphological alterations in the TM. Calculations suggest that the IOP effect seen in NT-treated mice was mostly or entirely explainable by changes in outflow facility, which together with our direct and indirect measurements of TM stiffness implicate the TM as the tissue mediating the effects of NT on IOP. Thus, this is the first demonstration of prevention and rescue from steroid-induced OHT by a 'TM-active' FDA-approved glaucoma medication and suggests that ROCKi compounds have effective anti-fibrotic activity in the TM.

Our results extend findings from an earlier report showing that a ROCKi lowered IOP in a transgenic mouse locally overexpressing connective tissue growth factor (CTGF) (*Junglas et al., 2012*), to a clinically relevant disease context by demonstrating NT activity in the conventional outflow pathway. Importantly, two significant phenotypic changes were observed in NT-treated eyes with OHT. The first was a rapid reversal of IOP elevation (within 1–2 days). The second was a significant decrease in accumulated ECM materials, namely FN and basal lamina material underlying the inner wall of SC. Both cellular contractility changes and ECM changes likely contribute to the observed decrease of TM tissue stiffness by NT. In addition to NT-mediated changes in ECM turnover, the observed changes in ECM composition and amount may also be due to NT-mediated opening of flow pathways and consequential removal of ECM. In any case, having a therapy for steroid-induced OHT that inhibits the cycle of fibrosis by restoring function to a diseased tissue offers a potential benefit for patients (*Figure 6*).

Evidence shows that the biomechanical properties of TM tissue play an important role in the regulation of outflow function and thus IOP (*Wang et al., 2017a*; *Wang et al., 2018*). For example, alterations in the biomechanical properties (i.e., stiffening) of the TM are observed in glaucomatous human donor eyes, which were hypothesized to be associated with dysregulation of the ECM (*Lütjen-Drecoll et al., 1986*; *Yue, 1996*; *Tamm and Fuchshofer, 2007*; *Keller et al., 2009*; *Tektas and Lütjen-Drecoll, 2009*; *Last et al., 2011*). TM stiffening and elevated IOP were also observed in animal models of steroid-induced OHT (*Raghunathan et al., 2015*; *Li et al., 2019*). In a recent study, we showed that TM tissue stiffness is associated with increased outflow resistance in steroid-induced OHT mouse eyes (*Wang et al., 2018*). We consider it noteworthy that NT appears to have restored normal TM biomechanical properties in our steroid-induced OHT model after only 4 days of

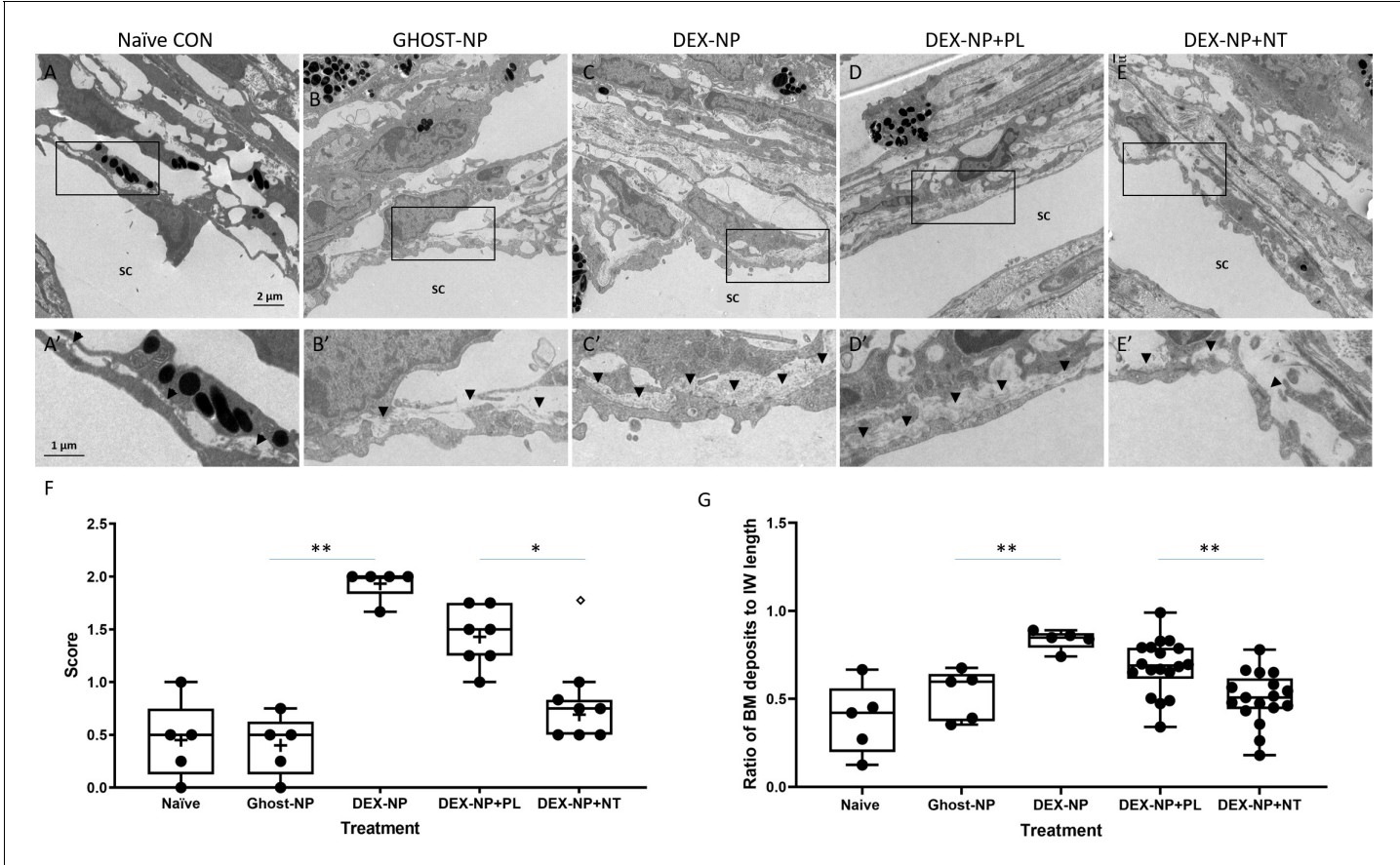

**Figure 5.** Netarsudil (NT) partially normalized ultrastructure of conventional outflow tissues in ocular hypertensive eyes. Two groups of age- and gender-matched wild-type C57BL/6 mice were bilaterally treated with dexamethasone-loaded nanoparticles (DEX-NPs) 1–2 times per week for 4 weeks. During the last week of DEX-NPs exposure, one group of mice was unilaterally treated with NT and another group with placebo (PL) once per day for 4 consecutive days. At day 5, eyes were collected and fixed with 4% PFA plus 1% glutaraldehyde in phosphate buffered saline (PBS) for 1–3 days at 4°C. The anterior segments were embedded in Epon, sectioned, stained with uranyl acetate/lead citrate, and examined with a JEM-1400 electron microscope. We show representative images from (**A**) five eyes naïve to NPs or treatment, (**B**) five control eyes (injected with NPs not loaded with DEX [Ghost-NP]), (**C**) five DEX-NP treated eyes, (**D**) 11 PL-treated eyes, and (**E**) nine NT-treated eyes. (**A'–E'**) show enlarged areas indicated by boxes in (**A–E**). Arrowheads point to basement membrane materials (BMM) underlying Schlemm's canal (SC) inner wall endothelial cells. (**F**) Summary of results from semi-quantitative scoring of extracellular matrix (ECM) in juxtacanalicular (JCT) region, paying particular attention to ECM underlying SC endothelial cells. Images in Panels A and B were scored as '0' (normal appearance), while the image in Panel C was scored as '2' (continuous and extensive ECM below SC and ECM in JCT). Shown are mean values for each eye, corresponding to 10, 13, 18, 12, and 8 images in each group that were graded. **p<0.01 comparing Ghost-NP vs. DEX-NP; *p<0.05 comparing DEX-NP+PL vs. DEX-NP+NT. The empty symbol was statistically determined to be an outlier of the DEX-NP+NT data set. (**G**) Summary of results from quantifying the amount of BMM underlying the inner wall of SC. The BMM ratio was calculated by measuring the length of the BMM underlying the SC inner wall and dividing the value by the SC inner wall's total length (***Figure 5—figure supplement 1***). Shown are mean values for each eye taken from 8, 10, 12, 34, and 33 images in each group. **p<0.01 comparing Ghost-NP vs. DEX-NP or DEX-NP+PL vs. DEX-NP+NT. Source data for semi-quantitative scoring of JCT and quantitative measurements of BMM in ***Figure 5—source data 1*** and ***Figure 5—source data 2***, respectively.

The online version of this article includes the following source data and figure supplement(s) for figure 5:

**Source data 1.** Source data for semi-quantitative scoring of JCT in ***Figure 5***.

**Source data 2.** Source data for quantitative measurements of BMM in ***Figure 5***.

**Figure supplement 1.** Quantification of basement membrane material (BMM) underlying the inner wall of Schlemm's canal.

treatment, and that this restoration was accompanied by significant morphological changes in the ECM of the TM. These in vivo observations in a relevant disease model confirm the importance of the rho-kinase pathway, and more generally the importance of cellular biomechanical tension as mediated through actomyosin cytoskeletal tension, on ECM synthesis, assembly, and degradation (***Rao et al., 2001***; ***Nakajima et al., 2005***; ***Pattabiraman and Rao, 2010***). The rapid timescale of the

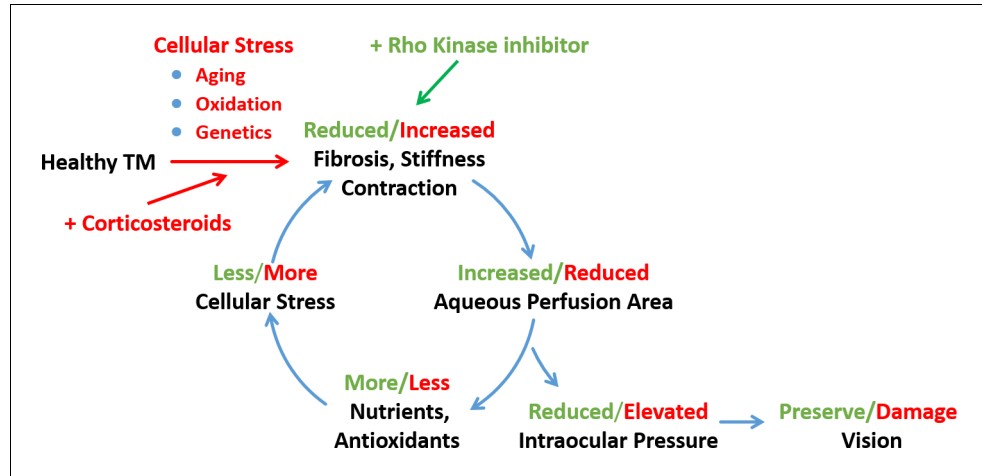

**Figure 6.** Schematic representation of a feed-forward model of fibrotic disease in the conventional outflow pathway responsible for ocular hypertension, incorporating a number of pathophysiological aspects of ocular hypertension (*Stamer and Acott, 2012*, *Schmidl et al., 2015*). The broader literature suggests that this feed-forward loop can be triggered by multiple factors, including aging, oxidative stress, or genetic predisposition. In this work we triggered this loop by corticosteroid administration and restored tissue function through rho-kinase inhibition. TM, trabecular meshwork.

observed ECM changes is consistent with observations from postmortem human eye preparations (*Keller et al., 2009*). The time course of structural changes in the TM was also consistent with the observed rapid (within 24 hr) pharmacodynamics of NT on IOP, significantly decreasing 1 and 3 months of OHT due to DEX treatment (*Figure 2* and *Figure 2—figure supplement 1*, respectively). This time course is consistent with known effects of ROCKis (*McMurtry et al., 2010*; *Lin et al., 2018*).

More generally, these data emphasize the anti-fibrotic properties of ROCKi in the conventional outflow pathway, consistent with a large body of literature in other tissues. Many mechanisms have been proposed for the systemically delivered anti-fibrotic effects of ROCKi, including inhibition of monocyte differentiation in a murine model of ischemic/reperfusion cardiomyopathy (*Haudek et al., 2009*), and macrophage infiltration of injured tissues in mouse models of multiple fibrotic diseases, including renal tubulointerstitial fibrosis, diabetic nephropathy, renal allograft rejection, peritoneal fibrosis, and atherosclerosis (*Wu et al., 2009*; *Knipe et al., 2015*). In view of the eye's immune-privi-leged status and local delivery of NT in the present study, it seems likely that resident cells of the conventional outflow pathway primarily mediate NT effects. In other tissues, RKIs inhibit TGF-β1-induced activation of p38 MAPK (*Holvoet et al., 2017*), activation of myocardin-related transcription factors (MRTFs), known to be master regulators of epithelial-mesenchymal transition (*Gasparics and Sebe, 2018*), and reduced activation of NF-κB (*Segain et al., 2003*). Importantly, downstream targets of MRTFs include CTGF and YAP/TAZ, and TGF-β-mediated signaling through p38 and NF-κB (*Raghunathan et al., 2013*; *Braunger et al., 2015*; *Inoue-Mochita et al., 2015*; *Tamm et al., 2015*; *Montecchi-Palmer et al., 2017*; *Hernandez et al., 2020*), all of which participate in the regulation of TM contractility, mechanotransduction, and IOP homeostasis. Thus, our DEX-induced OHT model represents a tool for further interrogation of these pathways in the context of an immune-privileged environment (*Figure 6*).

We used two complementary methods to estimate conventional outflow tissue stiffness, and each has advantages and disadvantages. AFM was used to directly measure the compressive mechanical properties of the conventional outflow tissues in mouse eyes at different locations of the TM and around the eye with high resolution (*Tao et al., 1992*; *A-Hassan et al., 1998*; *Matzke et al., 2001*; *Wang et al., 2017b*). However, AFM measurements of stiffness could only be conducted ex vivo on 'dead' tissues using a cryosectioning technique coupled with AFM. Thus, these measurements likely reflect the stiffness of the ECM, not cells (*Wang et al., 2017b*). In contrast, tensile stiffness estimates of conventional outflow tissues were derived from changes in area of SC lumen in living mouse eyes exposed to sequential IOP challenges. TM behavior was captured using spectral domain (SD)-OCT,

and stiffness was then quantified by iFEM as described previously (*Li et al., 2019*). The disadvantage with OCT is that eyes are imaged at only one location. Regardless, both methods showed consistent results, that is, that NT significantly reduced steroid-induced conventional outflow tissue stiffness. While desirable, technical concerns prevented OCT, AFM, and outflow facility measurements in the same eye.

Before the approval of NT in December 2017, four classes of glaucoma drugs (beta-adrenergic receptor antagonists, alpha-adrenergic agonists, carbonic anhydrase inhibitors, and/or prostaglandin analogues) were used to treat steroid-induced OHT. Unfortunately, none of these drugs directly target tissues responsible for homeostatic regulation of IOP in healthy eyes nor for the dysregulation of IOP in steroid-induced glaucoma. Instead, these drugs either decrease aqueous humor formation or divert aqueous humor from the conventional outflow pathway by increasing unconventional outflow drainage (*Bucolo et al., 2015*; *Schmidl et al., 2015*). Despite modest effects on conventional outflow (*Brubaker et al., 2001*; *Wan et al., 2007*; *Bahler et al., 2008*), prostaglandins are not often used to treat steroid-induced OHT due to concern about their effects on the blood-retinal barrier in vulnerable retinas (*Table 1*, *Moroi et al., 1999*). NT was the first agent to selectively target and modify conventional outflow morphology and function (*Li et al., 2016*; *Ren et al., 2016*). However, the current standard of care is for patients to be first treated with glaucoma medications that do not target the conventional outflow pathway, with NT being used only when these drugs are ineffective at IOP lowering. In two independent patient populations, we found 27 patients that had steroid-induced OHT and were treated with NT. Despite being first treated with multiple first- and second-line glaucoma medications (mean of 2.7 and 2.4), NT lowered IOP by an average of 7.9 mmHg in the first cohort and 6.0 mmHg in the second. These observations are consistent with mechanistic data from our mouse model, supporting the concept that the TM is the location of pathology in steroid-induced OHT, and emphasizing the importance of targeting the conventional outflow pathway in this well-recognized condition.

While patient data were encouraging, there were limitations to our approach. We used a retrospective chart review involving few patients that were treated with NT as a last resort by some, but not all physicians. Thus there was not a control group to determine how many patients were effectively managed on first- and/or second-line medications. These limitations along with the new data presented herein motivate the need for a randomized prospective clinical study to compare NT with first-line anti-glaucoma drugs in lowering steroid-induced OHT.

Steroid-induced OHT in mice was generated using nanotechnology (*Agrahari et al., 2017*), resulting in a model that closely matched steroid-induced observations in the human condition (decreased outflow facility, increased accumulation of ECM in the TM, and elevated IOP) (*Johnson et al., 1990*; *Johnson and Knepper, 1994*; *Johnson et al., 1997*; *Clark et al., 2001*). The present study suggests the utility of our mouse model of steroid-induced OHT and its translational relevance to the human clinical condition, where there is a high incidence of OHT in patients receiving intravitreal GC delivery implants to treat uveitis or macular edema. Our data indicate that human steroid-induced OHT patients who are refractory to standard glaucoma medications respond very well to NT. This retrospective study encourages a prospective study testing NT as a first-line drug for steroid-induced OHT. Further, our study utilizes methodology for non-contact, non-invasive estimation of conventional outflow function/health using OCT and iFEM. Our results confirm that this technology may have the resolution to detect changes in conventional outflow dysfunction due to GC treatment and restoration of function after drug treatment. In clinical practice, we suggest that this technology may have potential utility in staging glaucoma status and in monitoring treatment.

## Materials and methods

**Key resources table**

| Reagent type (species) or resource | Designation | Source or reference | Identifiers | Additional information |
|---|---|---|---|---|
| Software, algorithm | Schlemm3_2 | PMID:30651311 | Version 3.0 | Schlemm's canal segmentation |
| Software, algorithm | FEBio | https://febio.org/ | Version 3.0 | Finite element modeling |

*Continued on next page*

*Continued*

| Reagent type (species) or resource | Designation | Source or reference | Identifiers | Additional information |
|---|---|---|---|---|
| Antibody | Anti-αSMA (rabbit polyclonal) | Abcam, Cambridge, MA | ab5694 | IF (1:100) |
| Antibody | Anti-FN (mouse monoclonal) | Santa Cruz, Dallas, TX | sc8422 | IF (1:50) |

## Study design

Experiments were designed to test the hypothesis that dysregulation of conventional outflow function caused by GCs can be mitigated by NT treatment. The hypothesis was tested using a retrospective review of patients who were refractory to standard glaucoma treatments and in a validated mouse model of steroid-induced OHT that we developed (*Li et al., 2019*).

Data from humans were obtained using an unbiased retrospective analysis of electronic medical records of all patients seen by ophthalmologists at Duke Eye Center and from a retrospective chart review of two private clinical practices. All steroid-induced glaucoma patients treated with NT were included if not complicated with secondary diagnosis as detailed below. IOPs from one patient were excluded from the data set because they were greater than 1.5 times a quartile different from 75th percentile.

Mouse studies were based on sustained subconjunctival/periocular delivery of steroid to induce OHT (*Li et al., 2019*). Cohort sizes were determined using power analysis of data generated previously with this model (*Li et al., 2019*) and treatment effects with NT (*Li et al., 2016*). Eyes of mice were randomized to receive either NT or PL, and experimentalists were masked to treatment type. Primary endpoints (IOP, outflow facility, Young's modulus of TM, outflow tissue behavior visualized by OCT following IOP challenges, outflow tissue morphology by transmission electron microscope [TEM], and fibrotic marker expression by IHC) were established prior to start of experiments and all data were included in the analyses.

## Patient ascertainment, IOP measurement, and treatment

We identified two cohorts of patients with steroid-induced OHT who had been treated with NT. The first cohort was drawn from patients seen at the Duke Eye Center, using IRB-approved access to patient data via the EPIC electronic medical record system using SlicerDicer software. Search criteria included 'netarsudil' and associated ICD-10 codes for 'steroid responder' and 'steroid glaucoma' (H40.041, H40.042, H40.043, T38.0 × 5A, H40.60 × 0, H40.61 × 0, H40.62 × 0, and H40.63 × 0) between January 1, 2018, and March 1, 2020. Twenty-one eyes of 19 patients were identified as having started NT due to steroid-associated IOP elevation. None of the patients were 'tapered' from their steroid during the first month of treatment.

The second cohort was based on a comprehensive retrospective chart review of patients seen by Dr. John Samples at two different clinic sites: The Eye Clinic in Portland, Oregon, and Olympia Eye Clinic in Olympia, Washington. Charts of all patients seen by Dr. Samples between November 3 and December 1, 2019, were reviewed. Patients were included in the study that: (1) had a diagnosis of steroid-induced glaucoma, (2) were uncontrolled on standard glaucoma medications, (3) were treated with NT, and (4) had no confounding diagnoses (exfoliation glaucoma, pigmentary glaucoma, active neovascularization, narrow/closed angles, previous glaucoma surgery).

IOP in both studies was measured by Goldmann applanation tonometry before starting NT and then within 1 month after QD NT treatment.

## Animals

C57BL/6 (C57) mice (both males and females, ages from 3 to 6 months) were used in the current study. The animals were handled in accordance with approved protocols (A020-16-02 and A001-19-01, Institutional Animal Care and Use Committee of Duke University) and in compliance with the Association for Research in Vision and Ophthalmology (ARVO) Statement for the Use of Animals in Ophthalmic and Vision Research. The mice were purchased from the Jackson Laboratory (Bar Harbor, ME), bred/housed in clear cages and kept in housing rooms at 21℃ with a 12 hr:12 hr light:dark cycle.

## OHT animal model

OHT in mice was created by injection of NPs entrapping dexamethasone (DEX-NPs) into either the subconjunctival or periocular spaces as described in previous publications (*Wang et al., 2018*; *Li et al., 2019*). Briefly, DEX-NPs were diluted in PBS to a final NP concentration of 50 µg/µl, vortexed for 10 min, and then sonicated for 10 min. Mice were anesthetized with 100 mg/10 mg/kg of ketamine/xylazine. 20 µl per eye of the DEX-NP suspension (containing 1 mg of NPs with ~23 µg of DEX) was slowly injected bilaterally into either the superior or inferior subconjunctival or periocular spaces using a 30-gauge needle with a Hamilton glass microsyringe (50 µl volume; Hamilton Company, Reno, NV). After withdrawing the needle, Neomycin plus Polymyxin B Sulfate antibiotic ointment was applied to the eye and mice recovered on a warm pad. The injection was conducted 1–2 times per week for 3–4 weeks. For the last 4 days of the reversal study, DEX ophthalmic ointment (0.1%, Sandox Inc Cat# NDC 61314-631-36) was applied topically to both eyes once a day. For DEX-NP control treatment, nanoparticles without DEX (Ghost-NP) were used to treat mice in the same way.

## Drug treatments

NT and PL eye drops were provided in de-identified dropper bottles by Aerie Pharmaceuticals, Inc. All mice received bilateral treatments of DEX-NP but were randomized as to whether they were given NT or PL. In the prevention study, mice were treated unilaterally with 0.04% NT or PL by subconjunctival injection of 10 µl of NT or PL 1 day prior to bilateral DEX-NP treatment as described above, followed by NT or PL delivery 2–3 times per week for the entire 4-week duration of DEX-NP delivery. In the reversal study, eyes received DEX-NPs bilaterally 1–2 times per week for 3 weeks, followed by an additional 4 days when DEX was given once per day as topical ointment after unilateral treatment of NT or PL by subconjunctival injection (10 µl). This treatment regimen avoided potential ocular surface inflammation and animal health issues associated with multiple injections in the same eye during the last week of the study.

## IOP measurements

Mice were anesthetized with ketamine (60 mg/kg) and xylazine (6 mg/kg). IOP was measured immediately upon cessation of movement (i.e., in light sleep) using rebound tonometry (TonoLab, Icare, Raleigh, NC) between 10 am and 1 pm (*Li et al., 2014a*; *Li et al., 2014b*; *Meng et al., 2016*; *Li et al., 2018*; *Li et al., 2019*). Each recorded IOP was the average of six measurements, giving a total of 36 rebounds from the same eye per recorded IOP value. IOP measurements were conducted twice per week.

## Outflow facility measurements

Outflow facility was measured by a technician (IN) who was masked as to the treatment group using the iPerfusion system as described in detail previously (*Li et al., 2016*; *Li et al., 2019*). Briefly, at the end of the 4 days of NT/PL treatment period in the reversal study and after 4 weeks of NT/PL treatment in the prevention study, mice were euthanized using isoflurane, and eyes were carefully enucleated and rapidly mounted on a stabilization platform located in the center of a perfusion chamber using a small amount of cyanoacrylate glue (Loctite, Westlake, OH). The perfusion chamber was filled with pre-warmed Dulbecco's phosphate-buffered saline with added 5.5 mM D-glucose (DBG), submerging the eyes and regulating temperature at 35°C. Two glass microneedles, backfilled with filtered DBG, were connected to the system. Using micromanipulators, one microneedle was inserted into each anterior chamber of paired eyes without contacting the irises. Both eyes were perfused at 9 mmHg for 30 min to allow acclimatization and stabilization, followed by perfusion at nine sequential pressure steps of 4.5, 6, 7.5, 9, 10.5, 12, 15, 18, and 21 mmHg. Poor quality steps and subsequent pressure steps were eliminated using established criteria (*Sherwood et al., 2016*). Stable flow rate ($Q$) and pressure ($P$) averaged over 4 min at each pressure step were used for data analysis to compute outflow facility (*Meng et al., 2016*; *Sherwood et al., 2016*; *Li et al., 2018*).

## Calculations relating outflow facility changes and IOP changes

We asked whether measured IOP differences between NT and PL cohorts were quantitatively consistent with measured differences in outflow facility between these cohorts. For this purpose, we

assumed that unconventional outflow rate and episcleral venous pressure were unaffected by NT, that is, were the same between NT- and PL-treated cohorts. We further assumed a 10–15% reduction in aqueous inflow rate due to NT (*Kazemi et al., 2018*). We then used the modified Goldmann's equation (*Brubaker, 2004*) to determine a predicted IOP in the NT-treated eyes and compared this value to the actually measured IOP in these eyes. In brief, Goldmann's equation states $Q = C\,(IOP - EVP) + F_U$, where $Q$ is aqueous inflow rate, $EVP$ is episcleral venous pressure, and $F_U$ is unconventional outflow rate. Under the stated assumptions, the predicted value of IOP in NT-treated eyes is $IOP^{NT}_{predicted} = -\frac{xQ^{PL}}{C^{NT}} + \frac{C^{PL}}{C^{NT}}\,(IOP^{PL} - EVP) + EVP$, where superscripts indicate whether the value refers to NT- or PL-treated eyes; $Q^{PL}$ is the aqueous inflow rate computed from the measured IOP, measured facility, and an assumed value of $EVP$; and $x$ represents the percent decrease in aqueous inflow rate due to NT, here taken as $x = 10 - 15\%$. We assumed a range of EVP values and computed the ratio of the IOP predicted by the above formula to the actual measured IOP value in NT-treated eyes. IOP values used in the calculation were the cohort means of the last measurement taken before sacrifice, as follows: in the prevention study, 17.4 and 25.0 mmHg for NT- and PL-treated eyes, respectively, and in the reversal study, 21.0 and 28.4 mmHg for NT- and PL-treated eyes, respectively.

## Optical coherence tomographic imaging

At the end of the 4-day NT/PL treatment period in the reversal study and after 4 weeks of NT/PL treatment in the prevention study, OCT imaging was conducted in living mice. In vivo imaging utilized an Envisu R2200 high-resolution SD-OCT system (Bioptigen Inc, Research Triangle Park, NC). We followed our previously established techniques to image iridocorneal angle structures in mice (*Li et al., 2014a*; *Li et al., 2014b*; *Boussommier-Calleja et al., 2015*; *Meng et al., 2016*; *Li et al., 2019*). Briefly, mice were anesthetized with ketamine (100 mg/kg)/xylazine (10 mg/kg) and maintained with ketamine (60 mg/kg) every 20 min by IP administration. While mice were secured in a custom-made platform, a single pulled glass microneedle filled with PBS was inserted into the anterior chamber of one eye. The microneedle was connected to both a manometric column to adjust IOP and a pressure transducer (Honeywell Corp., Morristown, NJ) to continuously monitor IOP levels using PowerLab software. The OCT imaging probe was aimed at the nasal or temporal limbus and the image was centered and focused on the SC lumen. While collecting images, mouse eyes were subjected to a series of IOP steps (10, 12, 15, 17, and 20 mmHg) by adjusting the height of the fluid reservoir. At each IOP step, a sequence of repeated OCT B-scans (each with 1000 A-scans spanning 0.5 mm in lateral length) from spatially close positions was captured, registered, and averaged to create a high signal-to-noise-ratio image from the iridocorneal angle region of each animal. The duration of each pressure step was ~1–2 min. Note that the cohort of mice used for OCT imaging was different than that used for facility measurement.

## Segmentation of OCT images

OCT B-scans of iridocorneal angle tissues were registered and segmented following established methods (*Li et al., 2016*; *Meng et al., 2016*; *Li et al., 2019*) using SchlemmSeg software, which includes two modules: Schlemm I and Schlemm II. Briefly, OCT B-scans were automatically registered using our custom Schlemm I software for SC segmentation. The Schlemm II software package was then used to differentiate SC from scleral vessels, which were automatically marked. If SC was seen to be connected to collector channels (CC), manual separation of SC from CC was required, and was based on the shape of SC and speckling in the images generated by blood cells or other reflectors contained in blood vessels (*Mariampillai et al., 2008*; *Huang et al., 2009*; *Hendargo et al., 2013*; *Li et al., 2014a*; *Poole et al., 2014*; *Meng et al., 2016*). The speckle variance OCT-angiography images were generated based on the speckling in SC and vessels as described in detail in previous publication (*Meng et al., 2016*). SC was easily differentiated from other vessels due to its size and location.

## Segmentation reproducibility

To test the reproducibility of the SC segmentation process, we evaluated both interobserver and intraobserver reproducibility. The segmentation of SC was independently performed by two individuals. The first observer (GL) conducted the experiments and made initial measurements, then

repeated the measurements 1–2 months after the first examination to determine intraobserver reproducibility. The second observer (JC) was first given a training set of images to evaluate, then reviewed the images for the present study in a masked fashion to assess the interobserver reproducibility.

## iFEM determination of TM stiffness

In brief, the FEM technique allows one to compute the deformation of a structure due to loads/forces; here we computed the deformation of irideocorneal angle tissues (including the TM) due to changing IOP. In our iFEM approach, the stiffness of the TM was parametrically varied until computed SC deformations matched experimental observations, at which point we concluded that the TM stiffness specified in the FEM approach matched the actual (in vivo) value, while masked to treatment group. A pseudo-2D FEM geometry generated in our previous publication (*Li et al., 2019*) was used to calculate TM tissue stiffness based on OCT images. We have previously shown (*Wang et al., 2017a*) that such a pseudo-3D approach does not yield results significantly different from a true 3D model, at a fraction of the workload. The model includes the TM, sclera/cornea, and the uvea, and is meshed with four-noded tetrahedral elements. Tissues are treated as incompressible, isotropic, and nonlinearly hyperelastic (incompressible Mooney-Rivlin material model). The TM is assigned a range of stiffnesses (20–240 kPa), and for each stiffness value, we simulate the deformation of irideocorneal angle tissues, thus determining the cross-sectional area of SC vs. IOP. The computed SC area is compared with experimental measurements, and the estimated TM stiffness is taken as the value that minimizes the least squares difference between the experimental and predicted normalized SC areas over the IOP range 10–20 mmHg. SC luminal pressure is estimated as previously described (*Li et al., 2019*; *Supplementary file 1*).

## AFM measurement of TM stiffness

TM stiffnesses were measured using a previously developed AFM technique on cryosections of mouse eyes (*Wang et al., 2017b*; *Wang et al., 2018*; *Li et al., 2019*). Briefly, de-identified eyes coated with optimal cutting temperature compound (O.C.T.; Tissue-Tek) were cryo-sectioned from three different quadrants on a Microm Cryostar NX70 cryostat (Dreieich, Germany). For each quadrant, a few 10-μm-thick sagittal cryosections were collected on adhesive slides (Plus Gold Slide, Electron Microscopy Sciences, Hatfield, PA) and stored for up to 30 min in ice-cold PBS prior to AFM analysis.

Samples were then transferred to an MFP-3D-Bio AFM (Asylum Research, Santa Barbara, CA) and kept continuously immersed in PBS during measurements at room temperature (*Figure 3—figure supplement 3*). TM stiffnesses were measured following the same protocol we used previously (*Wang et al., 2017b*). Specifically, cantilever probes were modified by attaching a spherical indenter of diameter 10 μm to smooth nanoscale variations in tissue mechanics. For each indentation, the indentation depth was 0.5–1 μm, with a maximum applied force of 7 nN and approach velocity of 8 μm/s. A Hertzian model was used to extract a Young's modulus stiffness value from the force vs. indentation curves. For each cryosection, the TM was first localized as the region between the pigmented ciliary body and the inner wall endothelium of SC. Multiple locations in the TM region (typically 3–9) were probed by the cantilever and three repeated measurements were conducted at each location. The average from the three measurements was taken as the TM stiffness at that location. TM stiffnesses from all locations within a cryosection were then averaged to obtain the TM stiffness of that cryosection, and values for cryosections within a quadrant were averaged to obtain the TM stiffness in that quadrant. The mean stiffness of all quadrants was taken as the TM stiffness of the eye. Typically, there were 135 force curves acquired per eye (three force curves per location, typically five locations per cryosection, and typically nine cryosections per eye). A small number of measured TM stiffness values were excluded in a post hoc analysis if there was disagreement from a second reviewer as to whether the measurement location lay within the TM.

## Histology, immunohistochemistry, and TEM

After OCT imaging, mice were decapitated under anesthesia, and eyes were collected and immersion fixed in 4% paraformaldehyde at 4°C overnight. The eyes were then bisected, and the posterior segments and lenses were removed. The anterior segments were cut into four quadrants. For

immunostaining, each quadrant was embedded into LR-White resin, and 1 µm sections were cut and immunostained with antibodies that specifically recognized either αSMA (1:100 dilution, rabbit poly-clonal, ab5694, Abcam, Cambridge, MA) or FN (1:50 dilution, mouse monoclonal, Santa Cruz, Dallas, TX). The secondary antibodies were peroxidase-conjugated AnffiniPure Goat Anti-Rabbit or mouse IgG H and L (Alexa Fluor 488; Jackson ImmunoResearch Laboratories, West Grove, PA) at 1:500 dilu-tion. All sections were processed at the same time. Images were captured using a Nikon Eclipse 90i confocal laser scanning microscope (Melville, NY). Images from NT- and PL-treated eyes were col-lected at identical intensity and gain settings on the same day (z stacks of nine 0.5 µm optical sec-tions for each image) (*Li et al., 2018*; *Li et al., 2019*). For electron microscopy studies, one quadrant per eye of the anterior segment was fixed in 2% glutaraldehyde and embedded in Epon resin and 65 nm sagittal thin sections were cut through iridocorneal tissues using an ultramicrotome (LEICA EM UC6, Leica Mikrosysteme GmbH, A-1170, Wien, Austria). Sections were stained with ura-nyl acetate/lead citrate and examined with a JEM-1400 electron microscope (JEOL USA, Peabody, MA).

## Quantitative analysis of FN and α-SMA

Using ImageJ software, the TM from each image was manually segmented (region of interest, ROI) in a masked fashion by one co-author (MK), and fluorescence intensity (mean grayscale value) within each ROI was determined (see *Figure 4—figure supplement 1* for example). The inner wall of SC provided the outer boundary of the ROI, while pigment from ciliary body structures served as the interior boundary. Only the filtering region of the TM (i.e., the region below SC) was analyzed.

## Quantification of ECM content in JCT

To quantify ECM in the JCT under the inner wall of SC, images were captured at 8000× magnifica-tion and masked as to the identity of the treatment group. Images were quantified in two ways. First, they were scored using a pre-established semi-quantitative scoring system by two individuals (WDS and CRE) with extensive experience in viewing TEM images of the conventional outflow pathway. Specifically, the density and extent of the ECM in the JCT region, in particular below the inner wall of SC, was scored on a scale of 0–2, with 0 representing 'normal' appearing basal lamina, 2 repre-senting continuous basement membrane materials (BMM) such as observed in the mouse steroid OHT model (*Overby et al., 2014*), and 1 representing conditions between the two extremes. Thus, images were included whereby the inner wall of SC was clearly visible, and images were of sufficient quality to examine its basal lamina.

The second quantification method followed an existing approach (*Overby et al., 2014*) wherein coverage of BMM immediately underlying the inner wall of SC was measured. In brief, the lengths of the continuous BMM in contact with the inner wall of SC and the total length of the inner wall were measured using ImageJ software as described previously in 97 TEM images from 50 mice (see *Fig-ure 5—figure supplement 1* for an example). Then, the sum of the continuous BMM length was divided by the total inner wall length to compute a ratio representing the percentage of the inner wall underlain by BMM. Only continuous BMM in contact with the inner wall was included in the BMM measurements, that is, regions where continuous BMM showed an optically clear offset from the inner wall were excluded. Ratio values from multiple images from the same eyes were averaged.

## Statistical analyses

Due to the fact that IOPs are normally distributed and the expected directional (lowering only) effect of NT on IOP, we analyzed patient IOP data using a paired t-test with one tail, assuming equal vari-ance. To analyze IOP measurements in mice, the time points from 3 to 28 days in the prevention study, or the time points from 1 to 4 days post-NT/PL delivery in the reversal study, were averaged from each eye to produce a single average IOP value per eye for subsequent data analysis. The Mann-Whitney U-test was used for analyzing significant difference between groups for IOP and OCT images. To analyze outflow facility measurements, we used the well-established fact that the under-lying distribution of outflow facility in mice is log-normally distributed (*Sherwood et al., 2016*). Thus, a weighted paired or unpaired (two-way) t-test (*Sherwood et al., 2016*) was applied to the log-transformed facilities. To analyze data quantifying fluorescence intensity, an F-test of difference among means from ANOVA was used, with a GEE (generalized estimating equations) approach

applied to account for multiple observations per mouse. To analyze BMM in mice, we used two different methods: a parametric approach and a non-parametric approach. In the parametric approach, an F-test of difference among means was used as part of an ANOVA, followed by pair-wise comparisons based on t-tests of differences between means. In the non-parametric analysis, a Kruskal-Wallis ANOVA was used to test for difference among medians, followed by pair-wise comparisons based on Wilcoxon rank sum tests for differences between medians. Data in box and whisker plots show median, 25th percentile, and 75th percentile (boxes), as well as minimum and maximum values (whiskers). Data in other plot formats are presented in the form of mean and 95% confidence interval or mean and standard error of the mean (SEM), as noted. A value of $p \leq 0.05$ was considered statistically significant.

## Acknowledgements

We thank Ying Hao (Duke Eye Center Core Facility), who prepared histology sections and helped with TEM. Dr. Vibhuti Agrahari helped with the preparation and characterization of the NPs. Dr. Sandra Stinnett performed statistical analysis of data quantifying basement membrane deposits and stiffness measurements. We acknowledge funding support from the BrightFocus Foundation, Clarksburg, MA; Research to Prevent Blindness Foundation, New York; the Georgia Research Alliance, Atlanta, GA; Aerie Pharmaceuticals, Durham, NC; and National Institutes of Health, Washington, DC (EY031710, EY030124, and EY005722). Aerie Pharmaceuticals had input into the choice of several standard endpoint measurements (IOP, outflow facility, smooth muscle actin, and FN expression), but did not have input into other measures, such as TEM (including quantification), OCT imaging and finite element modeling, AFM, and quantification of immunolabeling results. Aerie Pharmaceuticals had no role in the conduct of the research, and other sponsors and funding organizations had no role in either the design or conduct of the research.

## Additional information

### Competing interests

Casey Kopczynski: Dr. Kopczynski is an employee of Aerie Pharmaceuticals. John Samples: Dr. Samples has participated in speaker's bureaus for Allergan, Aerie, Bausch and Lomb, Novartis, Santen, Akorn, Nicox and Roche. Pratap Challa: Dr. Challa currently owns equity in Aerie Pharmaceuticals. C Ross Ethier: Dr. Ethier has received consulting fees from Equinox and EyeD Pharma. W Daniel Stamer: Dr. Stamer has received research support from Aerie Pharmaceuticals, Aerpio, Broadwing Bio, Editas Medicine, Bausch, Allergan and Regeneron; currently serves on scientific advisory boards for Qlaris, Aerpio, Glauconix and Broadwing Bio, and has stock options in Glauconix and Qlaris. The other authors declare that no competing interests exist.

### Funding

| Funder | Grant reference number | Author |
|---|---|---|
| National Institutes of Health | EY030124 | Sina Farsiu<br>W Daniel Stamer |
| National Institutes of Health | EY031710 | C Ross Ethier<br>W Daniel Stamer |
| National Institutes of Health | EY005722 | W Daniel Stamer |
| BrightFocus Foundation | | Guorong Li |
| Research to Prevent Blindness | | W Daniel Stamer |
| Georgia Research Alliance | | C Ross Ethier |
| Aerie Pharmaceuticals | | W Daniel Stamer |

The funders had no role in study design, data collection and interpretation, or the decision to submit the work for publication.

## Author contributions

Guorong Li, Conceptualization, Data curation, Formal analysis, Validation, Investigation, Visualization, Methodology, Writing - original draft; Chanyoung Lee, A Thomas Read, Ke Wang, Katherine Young, Data curation, Formal analysis, Validation, Investigation, Methodology; Jungmin Ha, Data curation, Formal analysis, Methodology, Writing - review and editing; Megan Kuhn, Data curation, Formal analysis, Investigation, Methodology, Writing - review and editing; Iris Navarro, Data curation, Formal analysis, Validation, Investigation, Visualization, Methodology; Jenny Cui, Formal analysis, Validation, Visualization; Rahul Gorijavolu, Data curation, Validation, Investigation; Todd Sulchek, Conceptualization, Resources, Supervision, Funding acquisition, Validation; Casey Kopczynski, Formal analysis, Writing - original draft; Sina Farsiu, Conceptualization, Resources, Software, Funding acquisition; John Samples, Data curation, Formal analysis, Validation, Investigation; Pratap Challa, Data curation, Formal analysis, Supervision, Validation, Investigation; C Ross Ethier, Conceptualization, Resources, Formal analysis, Supervision, Funding acquisition, Investigation, Writing - original draft, Project administration, Writing - review and editing; W Daniel Stamer, Conceptualization, Resources, Formal analysis, Supervision, Funding acquisition, Validation, Investigation, Visualization, Methodology, Writing - original draft, Project administration, Writing - review and editing

## Author ORCIDs

Casey Kopczynski (i) http://orcid.org/0000-0002-2173-8309
W Daniel Stamer (i) https://orcid.org/0000-0002-2504-8997

## Ethics

Human subjects: Of two cohorts of de-identified patients, the first was drawn from patients seen at the Duke Eye Center, using IRB-approved access to patient data via the EPIC electronic medical record system using SlicerDicer software. The second cohort was derived from comprehensive retrospective chart review of patients at two different clinic sites: The Eye Clinic in Portland, Oregon and Olympia Eye Clinic in Olympia, Washington.

Animal experimentation: The mice were handled in accordance with approved protocols (A020-16-02 and A001-19-01, Institutional Animal Care and Use Committee of Duke University) and in compliance with the Association for Research in Vision and Ophthalmology (ARVO) Statement for the Use of Animals in Ophthalmic and Vision Research.

## Decision letter and Author response

Decision letter https://doi.org/10.7554/eLife.60831.sa1
Author response https://doi.org/10.7554/eLife.60831.sa2

# Additional files

## Supplementary files

• Supplementary file 1. Estimated pressures within Schlemm's canal (SC) lumen as a function of clamped intraocular pressure (IOP) levels using two-series resistor model of conventional outflow pathway described previously.

• Transparent reporting form

## Data availability

All data generated or analysed during this study are included in the manuscript and supporting files. Source data files have been provided for Figures 1, 2, 3, 4 and 5.

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
