## [Decision Letter]

**Acceptance summary:**

Steroid induced increased intraocular pressure is a significant problem in many eye diseases. Li et al. report that in a small retrospective study netarsudil, a rho-kinase inhibitor, rapidly reversed glucocorticoid-induced ocular hypertension in patients whose intraocular pressures were not controlled by standard medications. The authors show in mechanistic studies in a mouse model of glucocorticoid-induced ocular hypertension that netarsudil both prevented and reversed intraocular pressure elevation. A prospective clinical trial evaluating rho kinase inhibitors versus standard intraocular pressure-lowering agents in steroid-induced ocular hypertension is warranted.

**Decision letter after peer review:**

Thank you for submitting your article "Antifibrotic activity of a rho-kinase inhibitor restores outflow function and intraocular pressure homeostasis" for consideration by *eLife*. Your article has been reviewed by three peer reviewers, and the evaluation has been overseen by a Reviewing Editor and a Senior Editor. The following individuals involved in review of your submission have agreed to reveal their identity: Colleen McDowell (Reviewer #1); Ernst Tamm (Reviewer #2).

The reviewers have discussed the reviews with one another and the Reviewing Editor has drafted this decision to help you prepare a revised submission.

Summary:

This paper showed the Rho-kinase inhibitor netarsudil reverses glucocorticoid-induced ocular hypertension in patients, and also investigated the potential mechanism behind this using steroid-induced ocular hypertension mouse model. The authors showed that netarsudil increased outflow facility, decreased mechanical stiffness of the TM, as well as reduced expression of α smooth muscle actin and fibronectin with IHC. The implications of the work have great clinical significance.

Essential revisions:

This is an important study that addresses a fundamental problem in glaucoma. In this manuscript the authors provide convincing evidence that a Rho-kinase inhibitor substantially lowers IOP in patients with steroid-induced glaucoma that was uncontrolled by other medications. In addition, they show that this is also the case in a mouse model of steroid-induced glaucoma, with some prior publications in the field. However, there are some major issues that should be addressed

Since this is about a commercially available drug and partially funded by the company that markets it, the results need to be watertight which they are currently not. It is important that claims not be exaggerated.

The claim that netarsudil "reverses" glaucoma is not justified in the absence of visual field testing and other functional assessments of the presence, and then absence, of glaucoma. The authors don't show that the mice or the humans are cured and all signs of their disease erased, which is what "reversal" would necessitate. The same principle applies to the word "rescue." The term "remarkable" should also be removed.

To claim the antifibrotic potential of netarsudil, the authors need more quantified data to support the "antifibrotic" conclusions.

Since mouse and human steroid-induced glaucoma is a form of glaucoma with a substantial increase in a specific form of extracellular matrix (basal lamina-like deposits), it is tempting to speculate that those deposits cause stiffness, low outflow facility and high IOP. It would be a major breakthrough, if it could be shown convincingly that the removal of those deposits by rho kinase inhibitors restores TM function. Critical for this point (the "antifibrotic activity") though is to provide convincing and quantified data. Clearly, structural analysis is the gold standard to show fibrosis or the lack thereof.

The authors do this by a combination of immunohistochemistry and transmission electron microscopy. It appears though that those studies do not account for the considerable intra-individual differences that the trabecular meshwork outflow pathways show in an individual mouse eye. There are regions of high flow and low flow, and there is a substantial amount of published data showing that these differences are based on structural variations in the outflow pathways. Even in a normal mouse eye, there are regions with less or more fibronectin, α SM-actin etc. The authors are strongly recommended to perform additional molecular analyses. Western blotting or RT-PCR using protein/RNA obtained from corneoscleral rings is what scientists usually do to support data obtained by immunohistochemistry, and to address the problem of regional differences. Since the main ultrastructural finding in steroid-induced glaucoma, also in the mouse eye, are "basal lamina deposits". Immunostaining for basal lamina molecules such as laminin, nidogen, type IV collagen needs to be done support the TEM data.

For transmission electron microscopy, one quadrant was processed. It appears that several ultrathin sections were cut from this quadrant. Taking into account the thickness of ultrathin sections (40 – 80 nm) even the analysis of multiple consecutive sections would only have visualized a very tiny portion of TM circumference. Those regional differences need to be taken into account. Clearly there are also quite substantial regional differences when it comes to basal lamina deposits and other forms of fibrillar extracellular matrix in the TM. This is not only true for the circumference, but also for the TM in anterior-posterior direction. For the human patient, the authors quote the work by Johnson, Gottanka et al. which is still important after more than 20 years because of its thorough quantitative analysis. The bars should not be lower in the mouse eye, especially when it comes to the possible identification of a major mechanism. Other less time intensive methods may be available to quantify deposits in all regions, as long as the results represent the variability seen in the mouse eye so the conclusions are reliable.

If using TEM, the authors need to perform a power analysis in the normal mouse TM that shows how many sectors etc., need to be investigated to quantify basal lamina deposits, open spaces and ruffling of inner wall cells taking into account regional differences and regions with high and low flow. Based on those results quantification needs to be done. This is time consuming, but definitely feasible and the only way to support the statement of "antifibrotic activity". A semiquantitative analysis without knowing for certain the variability in the system is not recommended.

---

## [Author Response]

Essential revisions:This is an important study that addresses a fundamental problem in glaucoma. In this manuscript the authors provide convincing evidence that a Rho-kinase inhibitor substantially lowers IOP in patients with steroid-induced glaucoma that was uncontrolled by other medications. In addition, they show that this is also the case in a mouse model of steroid-induced glaucoma, with some prior publications in the field. However, there are some major issues that should be addressedSince this is about a commercially available drug and partially funded by the company that markets it, the results need to be watertight which they are currently not. It is important that claims not be exaggerated.

We agree with this assessment and have conducted further experiments (as directed below), and have adjusted language regarding claims (as directed below).

The claim that netarsudil "reverses" glaucoma is not justified in the absence of visual field testing and other functional assessments of the presence, and then absence, of glaucoma. The authors don't show that the mice or the humans are cured and all signs of their disease erased, which is what "reversal" would necessitate. The same principle applies to the word "rescue." The term "remarkable" should also be removed.

We agree that we have not shown that netarsudil reverses or rescues glaucoma. In most places, we were careful in our writing to say netarsudil reversed or rescued steroid-induced ocular hypertension, or features of steroid-induced ocular hypertension. However, there are several instances where we referred to our steroid-induced “glaucoma” model instead of the more correct “ocular hypertension” model.

We agree that reversal implies total recovery, which our ultrastructural data clearly shows is not the case. However, NT treatment did bring all of the other endpoint measurements to, or very near to, control levels. Regardless, in response to the reviewer’s concern we have softened our language, both in terms of reversal and other exaggerated claims where appropriate.

As recommended, we have replaced “remarkable” with “noteworthy”.

To claim the antifibrotic potential of netarsudil, the authors need more quantified data to support the "antifibrotic" conclusions.Since mouse and human steroid-induced glaucoma is a form of glaucoma with a substantial increase in a specific form of extracellular matrix (basal lamina-like deposits), it is tempting to speculate that those deposits cause stiffness, low outflow facility and high IOP. It would be a major breakthrough, if it could be shown convincingly that the removal of those deposits by rho kinase inhibitors restores TM function. Critical for this point (the "antifibrotic activity") though is to provide convincing and quantified data. Clearly, structural analysis is the gold standard to show fibrosis or the lack thereof.The authors do this by a combination of immunohistochemistry and transmission electron microscopy. It appears though that those studies do not account for the considerable intra-individual differences that the trabecular meshwork outflow pathways show in an individual mouse eye. There are regions of high flow and low flow, and there is a substantial amount of published data showing that these differences are based on structural variations in the outflow pathways. Even in a normal mouse eye, there are regions with less or more fibronectin, α SM-actin etc. The authors are strongly recommended to perform additional molecular analyses. Western blotting or RT-PCR using protein/RNA obtained from corneoscleral rings is what scientists usually do to support data obtained by immunohistochemistry, and to address the problem of regional differences. Since the main ultrastructural finding in steroid-induced glaucoma, also in the mouse eye, are "basal lamina deposits". Immunostaining for basal lamina molecules such as laminin, nidogen, type IV collagen needs to be done support the TEM data.For transmission electron microscopy, one quadrant was processed. It appears that several ultrathin sections were cut from this quadrant. Taking into account the thickness of ultrathin sections (40 – 80 nm) even the analysis of multiple consecutive sections would only have visualized a very tiny portion of TM circumference. Those regional differences need to be taken into account. Clearly there are also quite substantial regional differences when it comes to basal lamina deposits and other forms of fibrillar extracellular matrix in the TM. This is not only true for the circumference, but also for the TM in anterior-posterior direction. For the human patient, the authors quote the work by Johnson, Gottanka et al. which is still important after more than 20 years because of its thorough quantitative analysis. The bars should not be lower in the mouse eye, especially when it comes to the possible identification of a major mechanism. Other less time intensive methods may be available to quantify deposits in all regions, as long as the results represent the variability seen in the mouse eye so the conclusions are reliable.If using TEM, the authors need to perform a power analysis in the normal mouse TM that shows how many sectors etc., need to be investigated to quantify basal lamina deposits, open spaces and ruffling of inner wall cells taking into account regional differences and regions with high and low flow. Based on those results quantification needs to be done. This is time consuming, but definitely feasible and the only way to support the statement of "antifibrotic activity". A semiquantitative analysis without knowing for certain the variability in the system is not recommended.

The reviewers make multiple excellent points.

Regarding cohort size/power analysis: We originally chose cohort sizes based upon our previous study (PMID: 25028360), where Prof. Lütjen-Drecoll performed the quantitative analysis of ECM coverage under the inner wall of SC. In that study, significant differences due to dexamethasone treatment were observed using cohorts of 5 control and 7 treated animals (1-3 sections analyzed/eye). In the present study, our cohort sizes (11 placebo and 9 netarsudil-treated mice, 1-3 sections analyzed/eye) were larger than our previous study, and thus we suggest that our sampling is sufficiently representative.

Regarding segmental variability: the reviewers of course are correct that there is significant segmental variability in the mouse outflow tract. Ideally, one would perfuse flow tracer to label high- and low-flow regions, and quantify ECM in each region. Due to experimental constraints, we were unable to do such experiments as part of this study, although this work is planned. Instead, we have taken the approach of sampling more eyes and more widely spaced sections per eye (see details below) in a masked manner. By so doing, we account for regional variation through appropriate statistical analysis.

Regarding immunostaining for basal lamina molecules: We agree with the reviewers that these data would complement the TEM study. However, there are several practical considerations that mitigate against this. First, in our experience, the available antibodies to extracellular matrix proteins often take extensive testing and optimization from multiple sources. Second, we are uncertain which components are being altered by DEX, and thus would have to screen different candidates. Since the paper already has 7 supplemental figures, 2 tables and 6 figures (most of which have multiple subpanels), we instead focused our time on improving the TEM analysis, as suggested by the reviewers. Importantly, TEM quantitation in the Overby et al. study (Overby et al., 2014) observed similar changes in BMM from *both* mouse and human eyes exposed to corticosteroids.

Regarding quantitative scoring of fibrosis: While our semi-quantitative assessment of morphological changes in the conventional outflow pathway at the EM level was scored by two experts in this area, we agree that the paper would be enhanced by quantitative data. In direct response to the reviewer’s request, we have therefore conducted a quantitative analysis of our immunolabelling images and of our TEM images. In the TEM data set, images came from two opposite quadrants of each eye; in several cases, eyes were underrepresented (due to poor quality sections that could not be scored previously), so we cut additional sections from those eyes, as well as cutting sections from additional eyes in the reversal cohort that were not analyzed in the initial semi-quantitative scoring scheme. Using this enlarged image set (97 images from 50 eyes), we went on to quantify basement membrane material following the Lütjen-Drecoll approach (PMID: 25028360). This new quantitative data has been added to Figure 5. As the reviewers can see, the quantification of basement membrane material beneath inner wall produced results that are very similar to our previous semi-quantitative analysis of JCT, and which are quantitatively consistent with the previous report of Overby and colleagues (PMID: 25028360). Importantly, this quantitative data, which was gathered in a fully masked manner, shows statistically significant differences that confirm that treatment with netarsudil partially restored normalcy to the ultrastructure of the subendothelial region. We have added text to the Results as follows:

“When examined at the electron microscopic level, we observed two major effects of NT treatment on steroid-induced OHT eyes (Figure 5). The first was a significant reduction in the amount and density of basement membrane materials (BMM) underlying the inner wall of SC. The second was an apparent increase in the number of “open spaces” in the TM of NT-treated eyes, particularly in the JCT region. These two changes were scored on a semi-quantitative scale, confirming observational impressions of NT treatment (Figure 5F, p=0.02). In addition, we quantified the amount of BMM below the inner wall of SC in each treatment group following the Lütjen-Drecoll approach (Figure 5—figure supplement 1) (Overby et al., 2014). Similar to results reported in Overby et al., we found that length of BMM underlying SC was significantly increased in mice after DEX treatment (Figure 5G, p<0.001). Indeed, our BMM ratio measurements in control and DEX-treated mice were quantitatively similar to those reported by Overby and colleagues. Most important, these data confirm significant differences between PL- and NT-treated eyes (Figure 5G, p=0.001), whereby NT appears to partially restore the ultrastructure of the subendothelial region (Figure 5).”

An example methodological image has been provided in the supplemental materials (Figure 5—figure supplement 1), and the methodology for quantitation was added to Materials and methods, as follows:

“Quantification of extracellular matrix content in JCT

To quantify ECM in the JCT under the inner wall of SC, images were captured at 8000× magnification and masked as to the identity of the treatment group. Images were quantified in two ways. […] Only continuous BMM in contact with the inner wall was included in the BMM measurements, i.e. regions where continuous BMM showed an optically clear offset from the inner wall were excluded. Ratio values from multiple images from the same eyes were averaged.”